# Towards Physically Executable 3D Gaussian for Embodied Navigation

**Bingchen Miao[1,2], Rong Wei[2,], Zhiqi Ge[1], Xiaoquan Sun[2,3], Shiqi Gao[1], Jingzhe Zhu[1],**
**Renhan Wang[2], Siliang Tang[1], Jun Xiao[1], Rui Tang[2], Juncheng Li[1]***
[1] Zhejiang University, [2] Manycore Tech Inc
[3] Huazhong University of Science and Technology

## Abstract

3D Gaussian Splatting (3DGS), a 3D representation method with photorealistic real-time rendering capabilities, is regarded as an effective tool for narrowing the sim-to-real gap. However, it lacks fine-grained semantics and physical executability for Visual-Language Navigation (VLN). To address this, we propose **SAGE-3D** (**S**emantically and Physically **A**ligned **G**aussian **E**nvironments for **3D** Navigation), a new paradigm that upgrades 3DGS into an executable, semantically and physically aligned environment. It comprises two components: **(1) Object-Centric Semantic Grounding**, which adds object-level fine-grained annotations to 3DGS; and **(2) Physics-Aware Execution Jointing**, which embeds collision objects into 3DGS and constructs rich physical interfaces. We release **InteriorGS**, containing 1K object-annotated 3DGS indoor scene data, and introduce **SAGE-Bench**, the first 3DGS-based VLN benchmark with 2M VLN data. Experiments show that 3DGS scene data is more difficult to converge, while exhibiting strong generalizability, improving baseline performance by 31% on the VLN-CE Unseen task. The code is available in https://sage-3d.github.io.

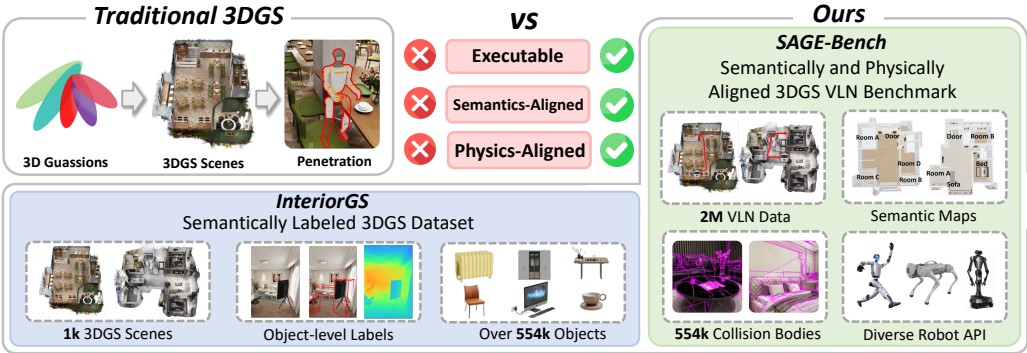

Figure 1: Traditional 3DGS vs. Our work. Compared with traditional 3DGS, our InteriorGS provides object-level 3DGS annotations across diverse indoor and outdoor scenes, including furnished homes, gyms, and concert halls as well as swimming pools and amusement parks. Meanwhile, SAGE-Bench contains semantically rich VLN data and detailed physical interfaces, representing a semantically and physically aligned 3DGS paradigm. Specifically, InteriorGS comprises 1,000 annotated 3DGS scenes with over 554k object instances, and SAGE-Bench, the first 3DGS-based VLN benchmark, features 2M trajectory-instruction pairs alongside a matching number of detailed collision bodies, laying a foundation for generalizable embodied navigation.

## 1 Introduction

Vision-and-Language Navigation (VLN) is a core capability for Vision-Language Action (VLA) models, enabling them to follow natural language instructions and navigate complex indoor spaces (Wei et al., 2025; Zhang et al., 2024). Direct real-world training is costly and risky, motivating the widely adopted sim-to-real paradigm (Qi et al., 2025; Zun Wang, 2023). Reducing the

---

*Corresponding author: junchengli@zju.edu.cn

Table 1: Comparisons with benchmarks for continuous navigation tasks. Here, "Instruction with Causality": tasks have causal dependencies rather than being mere "A-to-B" navigation; "Scene Geometry": whether the scene mesh is an imperfect estimate or accurate ground truth.

| Benchmarks | Num. of Task | Num. of Scenes | Scene Source | Instruction with Casuality | Scene Geometry | 3D Representation |
|---|---|---|---|---|---|---|
| VLN-CE (Krantz et al., 2020) | 4.5k | 90 | MP3D | ✗ | Estimated | Scanned Mesh |
| OVON (Yokoyama et al., 2024) | 53k | 181 | HM3D | ✗ | Estimated | Scanned Mesh |
| GOAT-Bench (Khanna* et al., 2024) | 725k | 181 | HM3D | ✗ | Estimated | Scanned Mesh |
| IR2R-CE (Krantz et al., 2022) | 414 | 71 | MP3D | ✗ | Estimated | Scanned Mesh |
| LHPR-VLN (Song et al., 2025) | 3.3k | 216 | HM3D | ✗ | Estimated | Scanned Mesh |
| OctoNav-Bench (Gao et al., 2025) | 45k | 438 | MP3D, HM3D | ✗ | Estimated | Scanned Mesh |
| **SAGE-Bench** | **2M** | **1000** | **InteriorGS** | ✓ | **Ground Truth** | **3DGS-Mesh Hybrid Representation** |

resulting sim-to-real gap has driven the evolution of scene representations, from early scanned mesh reconstructions such as Matterport3D (Chang et al., 2017) and HM3D (Ramakrishnan et al., 2021), to most recently 3D Gaussian Splatting (3DGS) (Kerbl et al., 2023).

Compared with prior VLN work (Krantz et al., 2020; Song et al., 2025) using scanned mesh reconstructions from RGB-D scans, 3DGS offers two key advantages: **1) Easier and more reliable object-level semantics.** Scanned mesh reconstructed from noisy depth scans forms a single continuous surface that merges objects into surrounding structures, making later separation costly (Cheng et al., 2025). In contrast, 3DGS represents scenes with discrete Gaussians that can be directly labeled. **2) View-consistent and photorealistic appearance.** Scanned mesh textures, stitched from sparse RGB viewpoints, often break under novel views, where incomplete coverage yields seams, stretching, or blur (Dalal et al., 2024). 3DGS instead optimizes a continuous radiance field, yielding consistent, photorealistic views from any navigable position—crucial for free-moving navigation.

Despite these advantages, the current 3DGS is solely used for high-fidelity rendering (Wang, 2024), as shown in the upper left corner of Fig. 1. It is unsuitable for effective application in VLN tasks due to its two significant limitations: **(1) 3DGS is deficient in fine-grained object-level semantics.** Existing 3DGS scenes contain only color and density information, with no instance IDs or object attributes (Li et al., 2024a). This makes it impossible to uniquely ground VLN instructions such as "go to the red chair next to the white bookshelf", and any attempt to recover object boundaries requires complex and error-prone post-processing. **(2) Lack of a physically executable structure.** Gaussian Splatting is, by nature, a volumetric rendering technique; although recent efforts (e.g., SuGaR (Guédon & Lepetit, 2024)) attempt to infer surface information from Gaussians, obtaining smooth surfaces remains challenging. Consequently, deriving reliable collision geometries from 3DGS is difficult, and aligning semantics with appearance is non-trivial.

In this work, we present **SAGE-3D (S**emantically and Physically **A**ligned **G**aussian **E**nvironments for **3D** Navigation), a paradigm that upgrades 3DGS **from a purely perceptual scene representation to an executable, semantically and physically aligned environment foundation** for embodied navigation. This transformation is enabled by two core components: **(1) Object-Level Semantic Grounding.** We sample 3DGS data from artist-created mesh scenes and create an object-level annotated indoor dataset through careful manual labeling and double verification, thereby endowing 3DGS with fine-grained semantics. Additionally, we design a 2D semantic top-down map derived from 3DGS to support instruction generation and path planning. **(2) Physics-Aware Execution Jointing.** We introduce a **3DGS-Mesh Hybrid Representation**: starting from our mesh scene data, we extract collision bodies for each object as the physics layer, while using 3DGS to provide photorealistic appearance. This decoupled design preserves high-fidelity rendering through 3DGS and enables accurate physical simulation based on mesh-based collision bodies, with connectivity to rich robotics APIs. Together, these two components transform 3DGS into a practical embodied navigation environment substrate and open new avenues for future embodied intelligence research.

Building on this, we release **InteriorGS**, a dataset consisting of 1,000 manually object-annotated 3DGS scenes. It covers diverse indoor and outdoor scenes, including furnished indoor environments, gyms and concert halls, as well as swimming pools and amusement parks, totaling over 554k object instances across 755 categories. We also introduce **SAGE-Bench** (Tab. 1), the first fully 3DGS-based VLN benchmark with 2M new trajectory-instruction pairs and a matching number of detailed collision bodies. **For data,** we provide a hierarchical instruction scheme that combines high-level semantic goals (especially task-causal ones like "I'm thirsty, get water from the table") with low-level actions (e.g., "move from stool to sofa"). **For evaluation,** we design three metrics

Figure 2: Overview of SAGE-3D, which consists of two key components: (1) Object-Level Semantic Grounding, 3DGS data is annotated by expect at the object level, then be transformed into 2D semantic maps for path planning and instruction generation; (2) Physics-Aware Execution Jointing, where scene and object collision bodies are generated via convex hull decomposition, integrated into 3DGS to form a 3DGS-Mesh Hybrid Representation, with extensive physics simulation interfaces.

for navigation natural continuity: Continuous Success Ratio, Integrated Collision Penalty, and Path Smoothness, to assess VLN models from the perspective of continuous motion.

Extensive experiments on SAGE-Bench yield several key insights: **(1) 3DGS scene data renders faster but is harder to converge than scanned mesh data.** 3DGS has a per-frame rendering time of 6.2ms, outperforming scaned mesh's 16.7ms. Yet reaching 40% Success Rate (SR) needs 160 iterations (6.2h) for 3DGS vs. 120 iterations (4.8h) for scaned mesh—this slower convergence stems from our 3DGS data's higher demands, as its richness and photorealism better mirror real-world complexity. **(2) Our scene-rich, photorealistic 3DGS VLN data exhibits strong generalizability.** Models trained entirely on this data achieve a significant performance improvement (31% SR increase) over baselines in unseen VLN-CE environments (Krantz et al., 2020), a result driven by the data's alignment with real-world scenarios. **(3) Our newly proposed three continuity metrics enable studying navigation's natural continuity, addressing gaps in conventional metrics.** Our newly designed navigation natural continuity metrics reveal that conventional metrics fail to capture model issues like continuous collisions and unsmooth motion, for example, in one experiment case, our ICP (indicating continuous collisions) reaches 0.87, while the traditional collision rate is only 1.

In summary, our contributions are as follows:

- We construct the first large-scale dataset of 1k fully furnished indoor 3DGS reconstructions with dense object-level annotations, released as **InteriorGS**.

- We propose **SAGE-3D**, a new paradigm that augments 3DGS with semantic granularity and physical validity, transforming it into an executable environment foundation.

- We build **SAGE-Bench**, a VLN benchmark based on 3DGS with fine-grained semantics, accurate per-object physical simulation, and rich interfaces for robot embodiments.

- We conduct extensive experiments based on our new paradigm and derive several novel insights in the VLN domain and validate the superiority of our newly introduced data.

## 2 SAGE-3D

In this section, we systematically introduce **SAGE-3D**, a novel embodied learning paradigm based on 3DGS, as illustrated in Fig. 2. We first provide a formal definition of this paradigm (Section 2.1), followed by an introduction add fine-grained semantic labels to 3DGS through manual annotation and the generation of 2D top-down semantic maps (Section 2.2). We then utilize convex hull decomposition to extract collision bodies and construct a rich physical simulation interface (Section 2.3).

## 2.1 SAGE-3D PARADIGM

We propose **SAGE-3D** (**S**emantically and Physically **A**ligned **G**aussian **E**nvironments for **3D** Navigation), a new paradigm that uses 3DGS as the environment foundation for training and evaluating embodied agent. This paradigm **upgrades 3DGS, originally used solely for photorealistic rendering, into an executable, semantically and physically aligned environment foundation** that supports continuous Vision-and-Language navigation and related tasks.

Formally, we define **SAGE-3D** as the process of transforming a Gaussian primitive set $G$ from a 3DGS scene, with added semantics $M$ and physics $\Phi$, into an executable environment:

$$G + M + \Phi \longrightarrow \mathcal{E}_{\text{exec}}$$

where $G = \{g_i\}_{i=1}^{N}$ is the set of Gaussian primitives, $M$ is the semantic layer (e.g., instance/category maps, attributes), and $\Phi$ is the physics layer (e.g., collision bodies, dynamics). The resulting environment can be formalized as a semantics- and physics-augmented POMDP (Partially Observable Markov Decision Process):

$$\mathcal{E} = (\mathcal{U}, \mathcal{S}, \mathcal{A}, \mathcal{O}, T, Z; M, \Phi),$$

where $\mathcal{U}$ is the instruction space, $\mathcal{S}$ the continuous state space, $\mathcal{A}$ the action space, $\mathcal{O}$ the multimodal observation space, and $T, Z$ are physics-driven state transition and rendering functions.

The core goal of this paradigm is to preserve the photorealistic rendering quality of 3DGS while introducing object-level semantics and physical executability, making 3DGS a viable environment foundation for training and evaluating embodied agents.

## 2.2 OBJECT-LEVEL SEMANTIC GROUNDING

Conventional 3DGS encodes appearance (e.g., color, density) but lacks instance IDs or object attributes, limiting precise object-level VLN instructions (Chen & Wang, 2025; Li et al., 2024a). To overcome this, we release InteriorGS, a manually annotated 3DGS dataset with object-level semantics, and introduce a 2D top-down semantic map generator to support instruction generation.

**InteriorGS.** We construct InteriorGS: a dataset of 1k high-fidelity 3DGS scenes across diverse indoor and outdoor environments. It consists of 752 residential indoor scenes and 248 public scenes, including indoor public spaces such as concert halls and gyms, as well as outdoor venues including amusement parks and swimming pools, with double-verified object-level annotations, including object categories, instance IDs, and bounding box information. The dataset contains over 554k object instances across 755 categories, providing a dense, semantically consistent, and broadly diverse foundation for training and evaluation.

InteriorGS's 3DGS data is sampled from our artist-created mesh scenes. To achieve reliable 3DGS reconstruction in occlusion-rich indoor environments, we render an average of 3,000 camera views per scene and use the open-source GSplat pipeline (Ye et al., 2025) to estimate the 3DGS parameters. The detailed sampling process is provided in Appendix B.

**2D Semantic Top-Down Map Generation.** Unlike scanned mesh workflows that build NavMesh (e.g., by exhaustive scene traversal in Habitat) (Song et al., 2025; Krantz et al., 2022), 3DGS lacks inherent semantics and discrete entities, making such representations infeasible. We therefore design **a 2D semantic top-down map** by projecting annotated 3D objects from InteriorGS onto the ground plane, with doors tagged by state (open / closed / half-open) and walls marked as non-traversable. Although annotations are stored as axis-aligned 3D boxes, we refine each footprint into an irregular mask by sampling object surface points, projecting them, and taking a 2D convex hull to optimize:

$$\mathcal{M}_k = \text{Fuse}\left(\text{Hull}\left\{\Pi_{\text{top}}(p) \mid p \in \text{Surf}(o_k)\right\}\right)$$

where $\mathcal{M}_k$ is the 2D mask for object $o_k$, $\text{Surf}(o_k)$ is the set of sampled surface points of object $o_k$, $\Pi_{\text{top}}$ is the projection onto the ground plane, $\text{Hull}(\cdot)$ denotes the 2D convex-hull operator, and $\text{Fuse}(\cdot)$ merges multi-view masks into a consistent footprint.

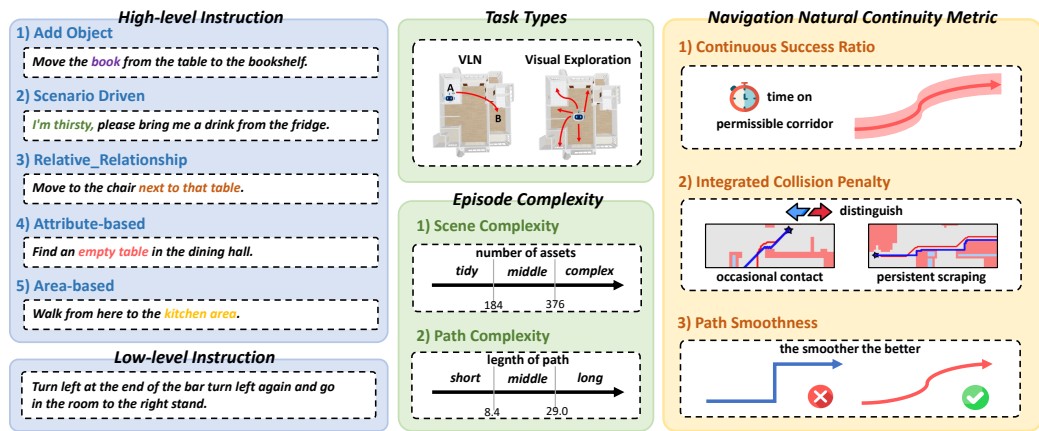

Figure 3: Overview of SAGE-Bench. SAGE-Bench includes a hierarchical instruction generation scheme, two major task types, two episode complexity categories, and three newly designed natural continuity metrics for navigation.

## 2.3 PHYSICS-AWARE EXECUTION JOINTING

3DGS with semantics still cannot serve directly as a VLN environment, as it allows issues such as mesh penetration that hinder embodied learning (Yue et al., 2024b). To overcome this, we extract object-level collision geometry, derive navigable space, and provide a physics simulation interface.

**Physics Simulation with 3DGS–Mesh Hybrid Representation.** Starting with version 5.0, Isaac Sim supports rendering 3DGS assets from USDZ files exported by 3DGUT (Wu et al., 2025a). However, the imported 3DGS are appearance-only and do not carry physics. To enable physically executable scenes, we take the artist-created triangle meshes of each object and apply CoACD (Wei et al., 2022) for convex decomposition, yielding per-object **collision bodies**. We then assemble a USDA scene where the collision bodies are authored as *invisible* rigid shapes (driving contact and dynamics), while the 3DGS file remain *visible* and provide photorealistic appearance. Concretely, each object is instantiated as a USD prim and augmented with $\Phi_k$ (rigid-body and contact parameters), where static-scene objects default to static bodies, and a curated subset is configured as movable or articulated to support extended interactions. This 3DGS–Mesh Hybrid Representation authoring removes the need to ray trace the artist meshes at runtime, preserves high-fidelity rendering through 3DGS, and supplies accurate collision geometry for physics.

**Agents, Control, and Observations in a Continuous Environment.** The simulator exposes robot APIs for legged and wheeled ground platforms (e.g., Unitree G1 / Go2 / H1) *and* aerial robots (e.g., quadrotor UAVs). Action interfaces support both discrete commands (e.g., `turn`/`forward`/`stop`) and continuous control—velocity commands $(v, \omega)$ for ground robots and 6-DoF velocity/attitude commands for UAVs—executed in a continuous environment (metric 3D space, no teleportation between panoramic nodes). The environment provides synchronized RGB, depth, semantic segmentation, poses, and contact events, along with built-in collision detection, stuck/interpenetration monitoring, and recovery. Offline-generated collision bodies are cached to accelerate loading and ensure stable, repeatable evaluation.

## 3 SAGE-BENCH

In this section, we introduce SAGE-Bench, the first 3DGS-based VLN benchmark, as shown in Fig. 3. It includes a hierarchical instruction generation scheme (Section 3.1), a three-axis evaluation framework (Section 3.2), and three navigation natural continuity metrics (Section 3.3).

### 3.1 DATA GENERATION

**Hierarchical Instruction Generation.** To address the limitations of current benchmarks (Zun Wang, 2023), particularly the lack of tasks with causal dependencies such as

"I'm thirsty, get water from the table", we introduce a hierarchical scheme that combines high-level semantics with low-level action primitives for more realistic navigation.

We define two levels of instructions: **High-level instructions** emphasize task semantics and human-oriented intent, and comprise 5 categories: *Add Object* (introducing causal objects or actions that make a trajectory contextually meaningful); *Scenario Driven* (embedding specific situational motives that make the destination a reasonable place for execution); *Relative Relationship* (distinguishing similar nearby targets via spatial relations such as "next to" or "opposite"); *Attribute-based* (identifying a unique target using perceivable attributes like color, state, or contents); *Area-based* (directing the agent toward a general functional area rather than a specific object). **Low-level instructions** focus on control and kinematic evaluation, including primitive actions such as forward moves. Detailed design and explanation can be found in Appendix C.

Low-level instructions are created by templating the start and end waypoints. High-level instructions are generated by feeding an MLLM with a prompt (detailed in Appendix C) constructed from object categories, attributes, and spatial relations in the 2D semantic map.

**Trajectory Generation.** Using the collision bodies from Section 2.1, we construct the final navigation map by combining a 1.2 m-height occupancy map with the 2D semantic map. Then we run A*-based shortest-path search to generate trajectories, more details can be found in Appendix A.

In total, we produce 2M new instruction–trajectory pairs for VLN. We balance the data distribution and select 1,148 samples to form the SAGE-Bench test split, including 944 high-level and 204 low-level samples across 35 distinct scenes, with the remainder used for training and validation.

## 3.2 THREE-AXIS EVALUATION FRAMEWORK

SAGE-Bench introduces a three-axis evaluation framework that orthogonally combines task types, instruction level, and episode complexity into discrete evaluation slices.

**Task Types.** This axis specifies the task paradigm and input form, considering two fundamental navigation tasks: VLN and Visual Exploration. Visual Exploration aims to drive the model to explore the environment as much as possible in order to test policy understanding of the environment and the safety of exploration. We select 100 scenes as the test set for Visual Exploration.

**Instruction Level.** This axis measures how semantic and structural complexity affects the model, and it is aligned with the hierarchical instruction generation scheme described in Section 3.1.

**Episode Complexity.** This axis quantifies task complexity, covering both scene complexity and path complexity. Scene complexity primarily refers to asset density: we define scenes with more than 376 assets as "many" and those with fewer than 184 assets as "few". Path complexity considers path length: we define paths longer than 29.0 m as "long" and those shorter than 8.4 m as "short".

## 3.3 NAVIGATION NATURAL CONTINUITY METRIC

As a new continuous navigation benchmark, to assess VLN model performance from the perspective of continuous motion, SAGE-Bench introduces three natural continuity metrics for navigation.

**Continuous Success Ratio (CSR).** It indicates the fraction of time the agent stays within a permissible corridor around the reference path. SR makes a 0/1 judgment only at the endpoint, whereas CSR measures the proportion of time the agent stays within a permissible corridor around the reference path while satisfying task conditions, thus reflecting "goal-consistent" behavior throughout the episode. Given a trajectory of length $T$, let $s(t) = \begin{cases} 1, & \text{pos}(t) \in \mathcal{C} \text{ and task conditions satisfied} \\ 0, & \text{otherwise} \end{cases}$ where $\mathcal{C}$ is defined by buffering the reference path with radius $r_{\text{tol}}$, then

$$\text{CSR} = \frac{1}{T} \sum_{t=1}^{T} s(t).$$

**Integrated Collision Penalty (ICP).** It measures the time-averaged collision intensity along the trajectory, capturing both the frequency and duration of contacts. Traditional collision rate (CR)

does not distinguish between occasional contact and persistent scraping. ICP integrates the collision intensity sequence $c(t) \in [0, 1]$ over time as a penalty:

$$\text{ICP} = \frac{1}{T} \sum_{t=1}^{T} c(t),$$

**Path Smoothness (PS).** It evaluates a normalized smoothness score derived from consecutive heading-change (or curvature) magnitudes, where higher values indicate smoother paths. Smoother paths reduce abrupt turns and acceleration changes, benefiting real robot feasibility and stable planning. PS is computed from the variance of consecutive heading changes:

$$\text{PS} = 1 - \frac{1}{T-1} \sum_{t=2}^{T} \min\left(\frac{|\Delta\theta_t|}{\pi}, 1\right), \quad \Delta\theta_t = \theta_t - \theta_{t-1},$$

Here $\theta_t$ denotes the agent's heading angle at trajectory time step $t$, and $\Delta\theta_t$ is the change in heading between two consecutive time steps.

## 4 EXPERIMENTS

### 4.1 EXPERIMENTAL SETUP

**Baseline.** Considering the current generalization capability of MLLM models (Teterwak et al., 2024; Li et al., 2025a; Huang et al., 2024; Zhou et al., 2025; Li et al., 2024b), we conducted evaluations on a wide range of models. **(1) Closed-source MLLMs as Agent**: Includes Qwen-VL-MAX (Bai et al., 2023), GPT-4.1, GPT-5. **(2) Open-source MLLMs as Agent**: Qwen2.5-VL-7B (Bai et al., 2023), InternVL-2.5-8B (Zhu et al., 2025a), InternVL-3-8B (Chen et al., 2024), Llama-3.2-11B. **(3) Vision-Language Models**: We selected VLN models that have been widely used in recent years, including NaviLLM (Zheng et al., 2024), NavGPT-2 (Zhou et al., 2024), CMA (Krantz et al., 2020), NaVid (Zhang et al., 2024), and NaVILA (Cheng et al., 2025).

**Evaluation Metric. (1) For the VLN task.** In addition to the three novel metrics we proposed in Section 3.3 for evaluating the natural continuity of model navigation — CSR, ICP, and PS — we also adopt common metrics used in VLN tasks, including success rate (SR), oracle success rate (OSR), and success weighted by path length (SPL) and Collision Rate (CR). **(2) For the Visual Exploration task.** There are two metrics: Episode Time and Explored Areas. An episode is terminated immediately if a collision occurs, and the maximum episode time is set to 120 seconds.

**Implementation Details.** We selected 500k "trajectory–instruction" pairs from SAGE-Bench, with no overlap with the test set. We trained two models on this subset: one based on NaVILA's pre-trained model navila-siglip-llama3-8b-v1.5-pretrain (denoted as NaVILA-base), producing NaVILA-SAGE; and the other based on Navid's pre-trained model navid-7b-full-224 (denoted as NaVid-base), producing NaVid-SAGE. Training details are shown in Appendix A.

### 4.2 RESULTS AND INSIGHTS

**Overall Comparison on SAGE-Bench.** Tab. 2 presents the experimental results of MLLMs and VLN models on SAGE-Bench. **(1) SAGE-Bench poses a novel and challenging VLN task for current VLN models and MLLMs.** Except for the recent SOTA VLN model NaVILA, other models achieve SR values no higher than 0.15. For instance, NaVid, which achieves 0.37 SR and 0.49 OSR on VLN-CE R2R Val-Unseen, only obtains 0.15 SR and 0.17 OSR on SAGE-Bench. Similarly, NaVILA, which achieves 0.54 SR and 0.63 OSR on VLN-CE R2R Val-Unseen, records only 0.39 SR and 0.47 OSR on SAGE-Bench. **(2) MLLMs' multimodal understanding inherently gives them some VLN capability.** Both the latest open-source and closed-source MLLMs achieve VLN SRs ranging from 0.10 to 0.14 on SAGE-Bench, comparable to dedicated VLN models such as CMA (0.13 SR) and NaVid (0.15 SR), and even surpass VLN models in OSR. For example, the 0.20 OSR achieved by InternVL-3 exceeds that of NaVid (0.17 OSR). **Notably**, several baseline models with weak VLN performance (SR < 0.20) fail to understand navigation instructions or environmental information in our challenging tasks, behaving like "random or single-action prediction" (e.g., continuous straight movement), rendering their CR, ICP, and PS metrics non-comparable.

Table 2: Comparison of different models on VLN and Visual Exploration tasks on SAGE-Bench. **Bold** values represent the best performance across all methods. Gray values indicate that these metrics lack comparative significance due to the low navigation performance of the models.

| Methods | VLN (High-level Instruction) | | | | | | | Visual Exploration | |
|---|---|---|---|---|---|---|---|---|---|
| | SR↑ | OSR↑ | SPL↑ | CR↓ | CSR↑ | ICP↓ | PS↑ | Episode Time↑ | Explored Areas↑ |
| *Closed-source MLLMs as Agent* | | | | | | | | | |
| Qwen-VL-MAX | 0.14 | 0.25 | 0.12 | 0.85 | 0.21 | 0.41 | 0.79 | 64.74 | 6.40 |
| GPT-4.1 | 0.13 | 0.21 | 0.12 | 0.72 | 0.19 | 0.35 | 0.81 | 67.70 | 3.00 |
| GPT-5 | 0.12 | 0.18 | 0.11 | 0.63 | 0.18 | 0.24 | 0.86 | 64.60 | 2.16 |
| *Open-source MLLMs as Agent* | | | | | | | | | |
| Qwen2.5-VL-7B | 0.13 | 0.14 | 0.13 | 0.71 | 0.21 | 0.27 | 0.87 | 42.19 | 6.88 |
| InternVL-2.5-8B | 0.10 | 0.13 | 0.10 | 0.52 | 0.14 | 0.33 | 0.88 | 28.82 | 4.28 |
| InternVL-3-8B | 0.12 | 0.20 | 0.11 | 0.64 | 0.17 | 0.32 | 0.82 | 34.70 | 6.34 |
| Llama-3.2-11B | 0.13 | 0.18 | 0.14 | 0.74 | 0.16 | 0.29 | 0.83 | 38.45 | 6.68 |
| *Vision-Language Model* | | | | | | | | | |
| NaviLLM | 0.05 | 0.06 | 0.05 | 0.21 | 0.09 | 0.24 | 0.90 | 18.73 | 5.74 |
| NavGPT-2 | 0.10 | 0.12 | 0.11 | 0.33 | 0.14 | 0.29 | 0.83 | 24.51 | 3.36 |
| CMA | 0.13 | 0.15 | 0.14 | 0.54 | 0.26 | 0.28 | 0.86 | 44.26 | 3.22 |
| NaVid | 0.15 | 0.17 | 0.15 | 1.24 | 0.29 | 0.33 | 0.89 | 56.13 | 4.28 |
| NaVILA | 0.39 | 0.47 | 0.34 | 3.28 | 0.48 | 0.61 | 0.68 | 77.82 | 8.40 |
| NaVid-base | 0.10 | 0.13 | 0.10 | 0.33 | 0.15 | 0.28 | 0.84 | 20.37 | 3.42 |
| **NaVid-SAGE (Ours)** | 0.36 | 0.46 | 0.32 | **2.12** | 0.48 | 0.66 | 0.54 | 60.35 | 5.66 |
| NaVILA-base | 0.21 | 0.26 | 0.22 | 3.53 | 0.33 | 0.72 | 0.41 | 58.26 | 6.52 |
| **NaVILA-SAGE (Ours)** | **0.46** | **0.55** | **0.48** | 2.67 | **0.57** | **0.54** | **0.74** | **82.48** | **8.74** |

Table 3: Rendering speed and training convergence comparison.

| Environment Type | Avg. Render Time / Frame (ms)↓ | Avg. Memory (MB)↓ | Iters to SR=40% (k)↓ | Time-to-SR=40% (hrs)↓ |
|---|---|---|---|---|
| Scanned Mesh (MP3D/HM3D) | 16.7 | 850 | 120 | 4.8 |
| 3DGS–Mesh Hybrid Representation (Ours) | 6.2 | 220 | 160 | 6.2 |

CR = 1  ICP = 0.87  PS = 0.42
*Case 1*

CR = 1  ICP = 0.65  PS = 0.53
*Case 2*

CR = 2  ICP = 0.74  PS = 0.71
*Case 3*

Figure 4: Visualization case study of navigation natural continuity. The red trajectory is the ground truth, and the blue Trajectory is the trajectory of NaVILA.

**Insight 1: 3DGS scene data renders faster than scanned mesh data but is harder to converge.** We randomly selected 10k training samples and 1k validation samples from both traditional scanned mesh data and our 3DGS data, and conducted experiments with the NaVILA-base model on an NVIDIA H20 GPU. Tab. 3 compares the rendering speed and model convergence between scanned mesh VLN data and our 3DGS VLN data. The results show that 3DGS scene data achieves a per-frame rendering time of 6.2 ms and an average memory usage of 220 MB, outperforming the 16.7 ms and 850 MB of scanned mesh data. However, in training, to reach the same 40% SR, the 3DGS-based model required about 160 iterations and 6.2 hours, while the scanned mesh-based model needed only about 120 iterations and 4.8 hours. This indicates that although 3DGS scene data offers faster rendering, it presents greater training difficulty and is relatively harder to converge.

**Insight 2: 3DGS scene data exhibits strong generalizability.** To evaluate the effectiveness of our novel 3DGS-based scene data, we tested the NaVILA-SAGE and NaVid-SAGE models, which were

Table 4: Results on VLN-CE.

| Methods | R2R Val-Unseen | | |
|---|---|---|---|
| | SR ↑ | OSR ↑ | SPL ↑ |
| Seq2Seq | 0.25 | 0.37 | 0.22 |
| Navid-base | 0.22 | 0.32 | 0.17 |
| **Navid-SAGE (Ours)** | 0.31 | 0.42 | 0.29 |
| CMA | 0.32 | 0.40 | 0.30 |
| NaVid | 0.37 | 0.49 | 0.36 |
| NaVILA-base | 0.29 | 0.38 | 0.27 |
| **NaVILA-SAGE (Ours)** | 0.38 | 0.51 | 0.36 |
| NaVILA | **0.50** | **0.58** | **0.45** |

Table 5: Results on different instruction levels.

| Methods | Instruction Level | SAGE-Bench VLN | | | | | |
|---|---|---|---|---|---|---|---|
| | | SR ↑ | OSR ↑ | SPL ↑ | CSR ↑ | ICP ↓ | PS ↑ |
| GPT-4.1 | Low-level | 0.22 | 0.37 | 0.19 | 0.27 | 0.60 | 0.70 |
| | High-level | 0.13 | 0.21 | 0.12 | 0.19 | 0.35 | 0.81 |
| InternVL-3-8B | Low-level | 0.20 | 0.35 | 0.18 | 0.26 | 0.61 | 0.69 |
| | High-level | 0.12 | 0.20 | 0.11 | 0.17 | 0.32 | 0.82 |
| NaVid | Low-level | 0.24 | 0.42 | 0.21 | 0.34 | 0.63 | 0.64 |
| | High-level | 0.15 | 0.17 | 0.15 | 0.29 | 0.33 | 0.89 |
| NaVILA | Low-level | 0.56 | 0.66 | 0.50 | 0.58 | 0.48 | 0.75 |
| | High-level | 0.39 | 0.47 | 0.34 | 0.48 | 0.61 | 0.68 |

Table 6: Impact of the number of scenes and samples on model performance.

| Data in # Train | | SAGE-Bench VLN | | | | | |
|---|---|---|---|---|---|---|---|
| #Scenes | #Samples | SR ↑ | OSR ↑ | SPL ↑ | CSR ↑ | ICP ↓ | PS ↑ |
| 800 | 240k | **0.42** | **0.47** | **0.42** | **0.50** | **0.61** | **0.63** |
| 800 | 120k | 0.40 | 0.43 | 0.40 | 0.48 | 0.62 | 0.62 |
| 800 | 60k | 0.36 | 0.42 | 0.38 | 0.46 | 0.64 | 0.58 |
| 400 | 120k | 0.34 | 0.39 | 0.35 | 0.44 | 0.67 | 0.54 |
| 400 | 60k | 0.31 | 0.37 | 0.33 | 0.43 | 0.67 | 0.52 |
| 400 | 30k | 0.28 | 0.35 | 0.31 | 0.43 | 0.69 | 0.49 |
| 400 | 15k | 0.25 | 0.31 | 0.27 | 0.39 | 0.70 | 0.46 |
| 200 | 60k | 0.27 | 0.33 | 0.29 | 0.41 | 0.70 | 0.47 |
| 100 | 60k | 0.23 | 0.29 | 0.26 | 0.38 | 0.71 | 0.44 |
| NaVILA-base | | 0.21 | 0.26 | 0.22 | 0.36 | 0.72 | 0.41 |

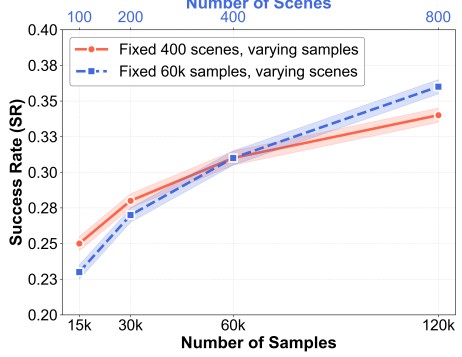

Figure 5: Model performance change curve (number of scenes vs. sample size).

trained solely on our SAGE-Bench dataset, on the VLN-CE benchmark. As shown in Tab. 4, models trained entirely on SAGE-Bench data (without any VLN-CE data) achieved clear performance improvements over their respective baselines. For example, NaVILA-SAGE achieved a 31% relative SR improvement on R2R Val-Unseen (from 0.29 to 0.38) and a 34% relative OSR improvement (from 0.38 to 0.51), with similar gains observed for the NaVid model.

**Insight 3: Our newly proposed three continuity metrics enable effective study of navigation's natural continuity, filling key gaps left by conventional metrics.** In Tab. 2, we report results for our three navigation natural continuity metrics. We observe that CSR is generally higher than SR, indicating a more inclusive and robust metric that does not require the model to fit the ground-truth trajectory exactly. For ICP and PS, although NaVILA attains relatively high task completion (0.39 SR, 0.47 OSR), it lacks natural motion continuity: an ICP of 0.61 indicates sustained collisions during navigation, and a PS of 0.68 reflects large, mechanical turning angles rather than smooth, natural motion. Additional visual examples in Fig. 4 corroborate this finding: the NaVILA model (blue trajectory) exhibits unsmooth movement and persistent collisions that conventional metrics fail to reveal. For instance, in Case 1, the model hugs the wall for a long period, yet the collision rate CR is only 1, while our ICP reaches 0.87.

## 4.3 More Findings

**High-level Instructions vs. Low-level Instructions.** Tab. 5 compares the performance of different models on high-level and low-level instructions in the VLN task. Compared with low-level instruction data, which are composed of basic step-by-step actions that guide the model gradually through the task, VLN models perform worse when executing high-level instructions. Even the recent SOTA model NaVILA achieves only a 0.39 success rate on high-level instructions, significantly lower than its 0.56 success rate on low-level instructions. Notably, high-level instructions, with their more natural semantics, are closer to real-life scenarios, presenting greater challenges for the future development of VLN models.

**Number of training Scenes vs. Training Sample Size.** Tab. 6 and Fig. 5 illustrate the influence of varying the number of scenes and the number of samples. We observe that increasing the number of scenes in the training data, while keeping the sample size constant, yields greater performance gains

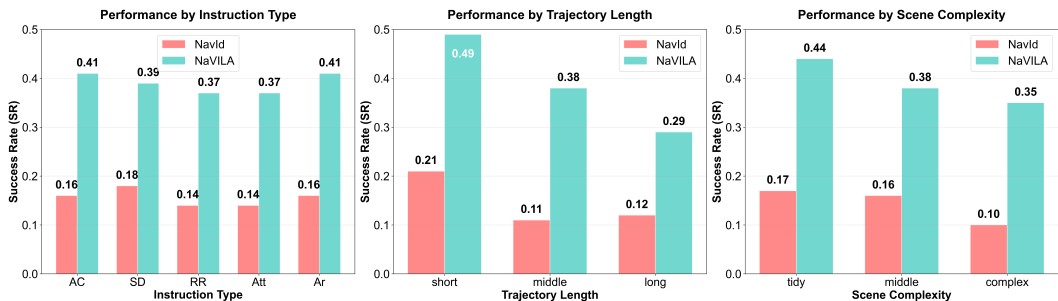

Figure 6: Results under Different Evaluation Slice.

than merely increasing the number of samples. Specifically, with the same number of augmented scenes (800), increasing the sampling density progressively improves the VLN model's performance on the val-unseen split. Conversely, generating the same number of samples (700k) from a larger number of environments produces better results. These findings indicate that the number of scenes (Scenes) has a greater impact than the number of samples (Samples), suggesting that diversity of environments is more critical for learning VLN.

**Results under Different Evaluation Slice.** Based on our three-axis evaluation framework, we further present experimental results in Fig. 6 for different high-level instruction types, trajectory lengths, and scene complexities. The results show that VLN models perform worse on the "Relative Relationship" and "Attribute-based" instruction types, with SR scores for both NaVILA and NaVid more than 2% lower than those for other types. In addition, as trajectory length increases and scene complexity grows, model performance drops significantly.

## 5 RELATED WORK

Vision-and-Language Navigation (VLN) was first introduced by (Anderson et al., 2018) on early Matterport3D-based discrete panoramic graphs, later extended to multilingual / longer-horizon settings by Ku et al. (2020) and remote object grounding by Qi et al. (2020); research shifted to continuous control with (Krantz et al., 2020) (VLN-CE) on Habitat (Savva et al., 2019), though mainstream benchmarks still rely on scan-mesh reconstructions (with texture/semantic limitations). 3D Gaussian Splatting (3DGS)—representing scenes efficiently via anisotropic Gaussian primitives for photorealistic real-time rendering—has been integrated into embodied learning, such as coupling with MuJoCo/Isaac Sim (Jia et al., 2025; Zhu et al., 2025b), adopting dual-representation (Gaussians for rendering, meshes for collision) (Lou et al., 2025; Wu et al., 2025b), and enhancing with lighting estimation (Phongthawee et al., 2024); however, native 3DGS lacks object-level semantics, needs cumbersome manual appearance/physics alignment, and struggles with precise VLN language grounding (Krantz et al., 2020; Savva et al., 2019).

## 6 CONCLUSION

We presented **SAGE-3D**, a paradigm that upgrades 3D Gaussian Splatting from a purely perceptual scene representation to an executable, semantically and physically aligned environment foundation for embodied navigation. We release **InteriorGS**, the first large-scale dataset of 1K fully furnished indoor 3DGS reconstructions with dense object-level annotations, which enables robust semantic grounding in photorealistic environments. By unifying InteriorGS with a physics-aware execution layer and a hierarchical instruction-evaluation benchmark, **SAGE-Bench**, our framework provides a coherent pipeline from high-fidelity data generation to physically valid evaluation. We expect SAGE-3D to serve as a foundation for future research in richer multi-step and semantic-aware navigation tasks, interactive manipulation, and broader sim-to-real studies.

## ACKNOWLEDGMENTS

This work was supported by the National Natural Science Foundation of China (62436007), National Key Research and Development Program of China (2025ZD0123100), the Zhejiang NSF (LQK26F020001), Key R&D Program of Zhejiang (2025C01001), Fundamental Research Funds for the Central Universities (226-2025-00057), Ningbo Yongjiang Talent Introduction Programme (2024A-401-G), Zhejiang University Education Foundation Qizhen Scholar Foundation.

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

# APPENDIX

## OVERVIEW

This is the Appendix for the paper "Towards Physically Executable 3D Gaussian for Embodied Navigation". In this supplementary material we present:

- The implementation details of trajectory generation and training are provided in Section A.

- The detailed InteriorGS data sampling and construction is described in Section B.

- The more specific explanation of the hierarchical instruction system and prompts are presented in Section C.

- The comparison of our 3DGS-Mesh Hybrid Representation data with traditional Matterport 3D data is illustrated in Section D.

- The more visualizations of InteriorGS scenes are shown in Section E.

- The visualization of the data distribution of our InteriorGS is presented in Section F.

## A    IMPLEMENTATION DETAILS

**Trajectory Generation.** We run A*-based shortest-path search to generate trajectories with a cost function that integrates free-space distance, narrow-passage penalties, and area preferences to ensure both obstacle avoidance and task feasibility. To diversify the dataset, start–end pairs are sampled across different rooms, functional areas, and object instances, and a minimum safety distance is enforced to avoid overly close viewpoints that would reduce the richness of the visual signal.

**Training.** We selected 500k "trajectory–instruction" pairs from SAGE-Bench, with no overlap with the test set. We trained two models on this subset: one based on NaVILA's pre-trained model navila-siglip-llama3-8b-v1.5-pretrain (denoted as NaVILA-base), producing NaVILA-SAGE; and the other based on Navid's pre-trained model navid-7b-full-224 (denoted as NaVid-base), producing NaVid-SAGE. Training was carried out on 8 NVIDIA Tesla H20 GPUs with a batch size of 256 and a learning rate of $2 \times 10^{-5}$. The training data did not include any VLN-CE R2R or RxR samples.

## B    DETAILED SAMPLING METHOD OF INTERIORGS

To obtain reliable 3D Gaussian Splatting (3DGS) reconstructions in occlusion-rich indoor settings, we render on average $\sim 3{,}000$ camera views per scene with a ray tracing renderer and estimate 3DGS parameters using the renderer-provided poses via the open-source `gsplat` pipeline. To mitigate undersampling, we employ two complementary camera placement policies:

**(1) Perimeter-aware floorplan sweeps ("surround").** For each room polygon $P$, we generate $m$ inwardly offset polygons $\{P^{(j)}\}_{j=1}^{m}$ according to a prescribed distance schedule, and allocate a global camera budget $n$ across polygons proportionally to their perimeters. Along each $P^{(j)}$, cameras are uniformly spaced with optical axes aligned to the inward edge normals. At every placement, we instantiate three tangential baselines (left / center / right) and three vertical tiers: *outer tiers*—lower at $150\,\mathrm{mm}$ above the floor pitched $+30°$ (up), middle at mid-height with $0°$ pitch, and upper at $500\,\mathrm{mm}$ below the ceiling pitched $-30°$ (down); *interior tiers* ($j > 1$)—heights are interpolated between the corresponding outer tiers, with upper tiers pitched $-15°$, lower tiers $+15°$, and the middle tier matching the outer middle.

**(2) Volume-uniform sampling.** We distribute the global camera budget across rooms in proportion to room volume to favor coverage in smaller compartments, then draw 3D positions via Poisson-disk sampling for space-filling uniformity. At each sampled position, six cameras with canonical yaw–pitch templates are instantiated, and a shared small random perturbation is applied to their orientations. Together, these policies emphasize inward-facing, depth-aware viewpoints that broaden coverage and reduce undersampling-induced 3DGS underfitting.

To select viewpoints at appropriate distances from mesh surfaces, Figure 7 presents the camera-sampling outcomes: viewpoints shown in green are retained as the final selections, whereas those in red are discarded for being too close to the nearest mesh surface (below a safety threshold).

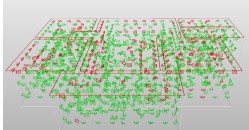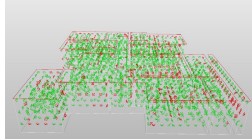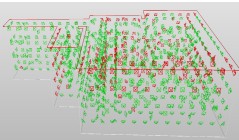

Figure 7: Camera pose sampling across four distinct floorplans. Green markers denote the final selected camera poses; red markers indicate poses discarded for being too close to the nearest mesh surface. Red outlines highlight ceiling–wall intersection regions, while white outlines indicate floor–wall intersections.

## C HIERARCHICAL INSTRUCTION GENERATION SCHEME

Grounded in 3DGS reconstructions and automatically generated 2D semantic top-down maps, we design a benchmarking-oriented hierarchical instruction generation scheme to close the gap left by prior VLN benchmarks that largely focus on low-semantic-granularity directives (e.g., "go from A to B" or atomic action sequences).

### C.1 HIGH-LEVEL INSTRUCTIONS: TASK-SEMANTIC ORIENTED

The most representative subset of High-Level Instructions is **Single-Goal Semantic Instructions**, which enriches basic "From A to B" navigation trajectories with semantic meaning. This subset addresses the limitation of traditional VLN benchmarks by linking navigation goals to human daily scenarios, object properties, or spatial relationships. Detailed categories and examples are provided below:

**(1) Add Object**  This category supplements a logical causal relationship between the start point and destination by introducing contextually relevant objects, making the navigation trajectory conform to human daily behavior. Without such causality, a directive like "from the sofa to the bookshelf" lacks practical meaning; adding a causal object (e.g., "books") transforms it into a goal-driven task.

- Case1: "Please move the book from the coffee table to the bookshelf in the study."
- Case2: "Please move the teacup from the coffee table to the bookshelf in the study."

**(2) Scenario Driven**  This category embeds a specific human-centric scenario or motive, framing the destination as a reasonable location to fulfill a practical need. The instruction directly reflects human intentions (e.g., thirst, hunger, rest), enabling the agent to associate navigation with task utility.

- Case1: "I'm thirsty, please bring me a drink from the fridge."
- Case2: "I want to rest, please take me to the sofa in the living room."

**(3) Relative Relationship**  This category defines the target using relative spatial terms to distinguish similar or adjacent objects—an essential capability for navigating cluttered environments (e.g., multiple chairs, tables). Common spatial terms include "next to," "behind," "the one on the left," "across from," and "in front of."

- Case1: "Move to the chair next to that table."
- Case2: "Walk to the cabinet across from the fridge in the kitchen."

**(4) Attribute-Based**  This category describes the target using perceivable, unique attributes to guide the agent in identifying a specific object among similar candidates. Attributes include color (e.g., "red"), state (e.g., "open," "on"), content (e.g., "empty," "full"), size (e.g., "large"), or decoration (e.g., "with a flower pattern").

- Case1: "Find an empty table in the dining hall."
- Case2: "Turn off the lit table lamp in the bedroom."

**(5) Area-Based** This category directs the agent to a general functional area rather than a specific object, focusing on spatial zones with practical purposes (e.g., cooking, resting, working). This is particularly useful for scenarios where the exact target object is unspecified but the functional context is clear.

- Case1: "Walk from here to the kitchen area."
- Case2: "Navigate to the lounge area in the living room."

## C.2 LOW-LEVEL INSTRUCTIONS: BASIC NAVIGATION & ACTION ORIENTED

Complementing the task-semantic focus of High-Level Instructions, Low-Level Instructions prioritize fundamental kinematic control and goal-directed point-to-point navigation without embedding complex contextual or functional semantics. These instructions serve two core purposes in our VLN framework: (1) evaluating an agent's basic motion execution capability (e.g., precise rotation, step control) and (2) providing a foundational navigation substrate for higher-level semantic tasks—acting as the "execution layer" that translates abstract High-Level goals into concrete movements. Unlike High-Level Instructions that answer "why to navigate," Low-Level Instructions focus solely on "how to move" or "where to go (without context)."

Below are the two primary categories of Low-Level Instructions, each tailored to assess distinct aspects of an agent's low-level navigation competence:

### C.2.1 1. BASE-ACTION: FUNDAMENTAL CONTROL BEHAVIORS

This category consists of goal-free primitive motions that test an agent's ability to execute basic locomotor or rotational commands with precision. These actions lack any spatial target (e.g., no specific object or area to reach) and instead focus on refining motion accuracy— a critical prerequisite for smooth, collision-free navigation in continuous environments. Common Base-Actions include step-based forward/backward movement and fixed-angle rotation.

- Case1:"Move forward two steps."
- Case2:"Turn 90 degrees to the right in place."
- Case3:"Turn 180 degrees to the left in place."
- Case4:"Move backward one step."

### C.2.2 2. SINGLE-GOAL: POINT-TO-POINT NAVIGATION

This category defines targeted point-to-point navigation tasks without additional semantic context—focusing solely on guiding the agent from a start location to a predefined end location. The end location can be a room, object, or functional zone, and the instruction is structured as a direct "go from X to Y" directive (or simplified to "go to Y" when the start location is implicit). This category is further subdivided based on the type of start and end targets, covering common indoor navigation scenarios:

- *Room-to-Room*: Navigate between two functional rooms. Case1:"Walk to the bedroom." Case2:"Go from the kitchen to the living room."
- *Room-to-Object*: Navigate from a room to a specific object within (or outside) the room. Case1:"Walk to the sofa in the living room." Case2:"Go from the study to the chair on the balcony."
- *Object-to-Object*: Navigate between two distinct objects. Case1:"Walk from the table to the door." Case2:"Go from the fridge to the dining table."
- *Object-to-Room*: Navigate from a specific object to a target room. Case1:"Go from the air conditioner to the kitchen." Case2:"Walk from the desk to the bedroom."

- *Zone-to-Zone*: Navigate between two functional sub-zones within a larger space. Case1:"Walk from the center of the kitchen to the sink area." Case2:"Go from the TV area in the living room to the window."

These Single-Goal Low-Level Instructions are critical for benchmarking an agent's spatial grounding ability (e.g., recognizing "bedroom" or "sofa" as navigation targets) without the confounding effects of semantic context, making them ideal for initial model training or control-focused evaluations.

## C.3 PROMPT FOR TRAJORIES TO INSTRUCTIONS

With the rapid advancement of MLLMs (Yue et al., 2024a; Li et al., 2026; Zeng et al., 2024; Sepehri et al., 2025; Li et al., 2023), leveraging them for data generation has become a widely adopted paradigm (Ye et al., 2024; Li et al., 2025b). Below, we present the prompt employed for instruction generation.

---

**Prompt for Instruction Generation**

```
You are a specialized data annotator for robotics.

Your mission is to act as a human providing natural language
instructions for a home or service robot. You will generate a diverse
set of human-centric navigation instructions (of INSTRUCTION TYPE
``High-Level-Deliver'') based on a symbolic TEXT MAP, STARTING POINT,
and END POINT.
You need to generate at least 2--4 instructions for each of the seven
INSTRUCTION TYPES defined below, ensuring variety and diversity.
```

⟨**Input**⟩

1. **TEXT MAP:** A textual description of an environment, including
   named areas, objects, and their unique IDs (e.g., Bar counter_0,
   chair_5). This map is the single source of truth.

2. **STARTING POINT:** The starting point of the trajectory,
   represented by an Object ID (e.g., chair_5).
   Example: ``starting_point'': ``chair_5''

3. **END POINT:** The endpoint of the trajectory, represented by an
   Object ID (e.g., sofa_0).
   Example: ``end_point'': ``sofa_0''

**<Task>**
```
Generate multiple natural language instructions for a trajectory from
the STARTING POINT to the END POINT (an optimal short path obtained
via A*). Use the TEXT MAP to understand the environment.
Generate at least 2--4 instructions for each of the INSTRUCTION TYPES
below, ensuring diversity.
```

**<Principles>**

1. **Don't Embellish or Exaggerate:** You do not know the intermediate
   path points or turns. Do not invent waypoints (e.g., \pass
   through desk_2") or directional commands (e.g., \turn left")
   unless explicitly stated in the map.

2. **NEVER Use Internal IDs:** Never include object IDs like chair_5.
   Instructions must be understandable to someone without the map.

3. **Stay Grounded in the Map:** Do not invent objects, properties, or
   spatial relationships not described or reasonably inferable from
   the TEXT MAP.

4. **Be Natural and Concise:** Use everyday language. Keep
   instructions between 5{20 words. Avoid robotic or overly formal
   phrasing.

---

5. **Be Creative and Diverse**: Vary sentence structure, vocabulary, and perspective. Small wording changes should yield meaningfully different instructions.

6. **Avoid Repetition**: Within each type, ensure instructions are semantically distinct|not just synonyms or minor rewordings.

7. **Ensure Executability**: Every instruction must be actionable using only the provided map.

8. **Strictly Adhere to Types**: Each instruction must clearly match its assigned type definition.

**<Instruction_Types>**

1. **Add_Object**
   *Description*: Adds a reasonable causality object (e.g., an object to carry) to justify the movement.
   *Examples*:

2. ``Please move the book from the coffee table to the bookshelf in the study.''

3. ``Please move the teacup from the coffee table to the bookshelf in the study.''

4. **Scenario_Driven**
   *Description*: Embeds the instruction in a human-centered scenario or goal.
   *Example*: ''I'm thirsty, please bring me a drink from the fridge.''

5. **Relative_Relationship**
   *Description*: Uses relative spatial terms (e.g., ``next to'', ''behind'', ``the one on the left'') to identify the target.
   *Example*: ''Move to the chair next to that table.''

6. **Attribute-based**
   *Description*: Describes the target using perceivable attributes (e.g., empty, with a vase, near a window).
   *Example*: ``Find an empty table in the dining hall.''

7. **Area-based**
   *Description*: Directs the robot to a functional area rather than a specific object.
   *Example*: ''Walk from here to the kitchen area.''

**<Output_Format>**
For each instruction type, generate 2--4 diverse instructions. Output as a JSON array:

```
[
  {
    ``instruction_type'': ``Add_Object'',
    ``start'': ``[provided_starting_object_id]'',
    ``end'': ``[provided_end_object_id]'',
    ``generated_instruction'': ``[instruction_text]''
  },
  {
    ``instruction_type'': ``Area-based'',
    ``trajectory_id'': ``[provided_id]'',
    ``start'': ``[provided_starting_object_id]'',
    ``end'': ``[provided_end_object_id]'',
    ``generated_instruction'': ``[instruction_text]''
  },
  ...
]
```

## D COMPARISON OF OUR DATA WITH MATTERPORT3D

In this section, we compare our 3DGS-Mesh Hybrid Representation with traditional Matterport3D data. As shown in Fig. 8, Matterport3D's mesh, derived from scanning, exhibits clear boundary ambiguity and object interpenetration, whereas our data uses collision bodies from the original mesh via convex decomposition, representing the ground truth. Rendered RGB images also show that our data is more photorealistic.

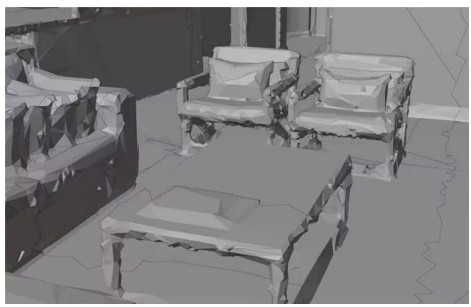
**Matterport 3D Estimated Mesh**

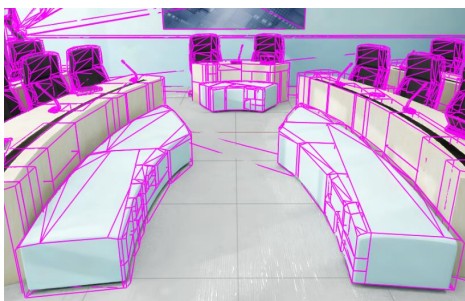
**Our Ground Truth Mesh**

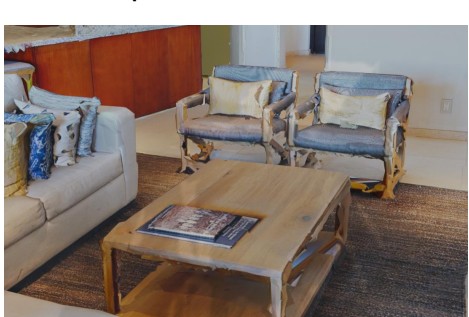
**Matterport 3D Rendering Scene**

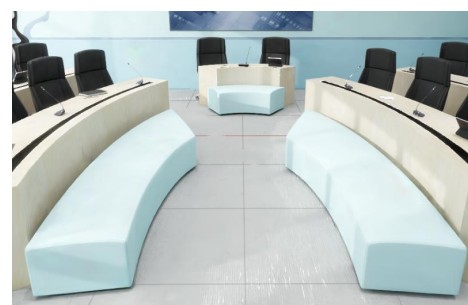
**Ours High-quality Rendering Scene**

Figure 8: Comparison of Our data with Matterport3D.

## E MORE VISUALIZATION OF INTERIORGS

This section presents additional InteriorGS scenes. As shown in Fig. 9, these scenes are highly detailed and photorealistic, demonstrating the high quality of our indoor data. We anticipate that InteriorGS will become a foundation for future embodied learning research.

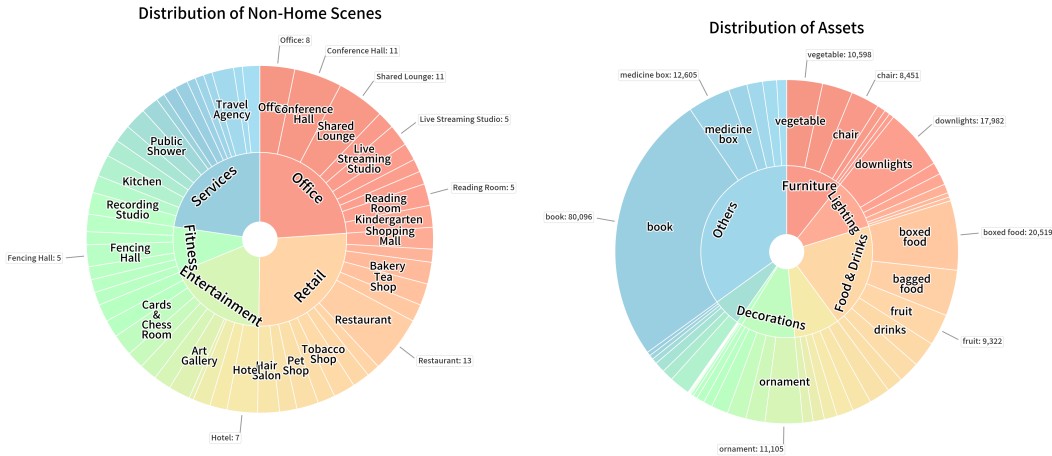

Figure 10: Distribution of non-home scenes of InteriorGS.

Figure 11: Distribution of assets of InteriorGS.

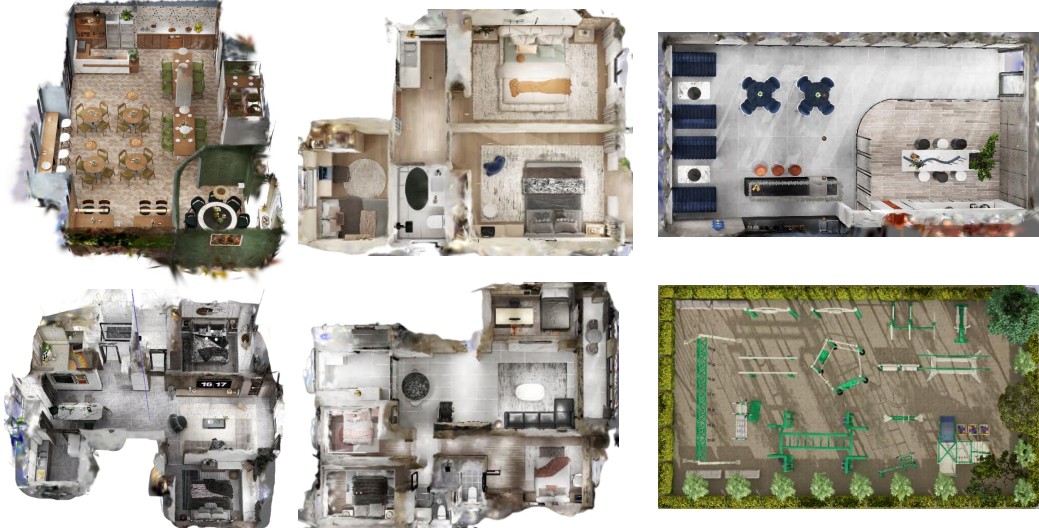

Figure 9: More Visualization of InteriorGS.

## F    DISTRIBUTION OF DATA FROM OUR INTERIORGS

In this section, we further detail InteriorGS's data distribution. Fig. 10 presents the distribution of 244 non-home scenes, categorized by function into Services, Office, Retail, Entertainment, and Fitness; Fitness has the fewest scenes, while the others are similarly distributed. Fig. 11 shows the asset distribution, including Furniture, Lighting, Food & Drinks, Daily Items, Decorations, and Others; books within Others are the most numerous assets.

