# OpenReview forum: "Towards Physically Executable 3D Gaussian for Embodied Navigation"
_ICLR.cc/2026/Conference — ICLR 2026 Poster_

### Official Review · Reviewer_zosw · 2025-10-20

**Soundness:** 4
**Presentation:** 3
**Contribution:** 3
**Rating:** 8
**Confidence:** 5

**Summary:**

This paper proposes a new paradigm named SAGE-3D (i.e., Semantically and Physically Aligned Gaussian Environments for 3D Navigation) that upgrades 3DGS into an executable, semantically and physically aligned environment. It contains two components, including Object-Centric Semantic Grounding and Physics-Aware Execution Jointing. It achieves faster scene rendering and better generalizability than scanned mesh data. Besides, this paper releases InteriorGS dataset containing 1000 manually object-annotated 3DGS scenes, and constructs SAGE-Bench including 2M new trajectory-instruction pairs and 554k detailed collision bodies. Extensive experiments demonstrate the effectiveness of the proposed paradigm.

**Strengths:**

Strengths:
1) The idea of SAGE-3D that upgrades 3DGS from a purely perceptual scene representation to an executable, semantically and physically aligned environment foundation is novel and valuable.
2) The constructed InteriorGS containing 1000 manually object-annotated 3DGS scenes, is beneficial for the field developement. The SAGE-Bench, the first fully 3DGS-based VLN benchmark with 2M new trajectory-instruction pairs and 554K detailed collision bodies, is beneficial for the downstream applications.
3) Extensive experiments are conducted on the proposed paradigm and validate the superiority of the newly introduced data.

**Weaknesses:**

Weaknesses:
1) In the second paragraph of Introduction section, the authors claims 3DGS offers three key advantages, but only two items are listed as advantages.
2) The organization of the scene data is based on GSplat, which may introduce more gaussian parameters than the mesh representation. Besides, the physics simulation introduces 3DGS-Mesh Hybrid representation, the collision bodies is computed by manually-created triangle mesh, does it introduce extra memory cost and rendering complexibility?
3) The data generation of SAGE-Bench is developed based on the IneriorGS, but is similar with the construction of existing VLN datasets. What is the core contribution of SAGE-Bench? It is not clear.
4) The definition of Nogoal-Nav is similar with visual exploration tasks, is there any difference? If there is no difference, there is no need to rename an existing task, visual exploration is more suitable.

**Questions:**

Please try to address the weaknesses.

---

> ### Author Response · Authors · 2025-11-26
> **To Reviewer zosw (Part 1)**
>
> We sincerely appreciate your insightful feedback. We are encouraged by your strong endorsement of our work in terms of Originality, Significance, and Clarity. We will explain your questions point by point.
>
> **Q1:**
>
> > In the second paragraph of Introduction section, the authors claims 3DGS offers three key advantages, but only two items are listed as advantages.
>
> **A1:** Thank you for your careful correction. This is an oversight in our writing process, which is a clear typographical error—we believe the core advantages of 3DGS **are actually the two items listed in the paper**. We mistakenly wrote "three key advantages", causing confusion in your understanding, and we sincerely apologize for that. We have corrected this in the revised version and thank you again for your correction.
>
>
>
> **Q2:**
>
> > The organization of the scene data is based on GSplat, which may introduce more gaussian parameters than the mesh representation. Besides, the physics simulation introduces 3DGS-Mesh Hybrid representation, the collision bodies is computed by manually-created triangle mesh, does it introduce extra memory cost and rendering complexibility?
>
> **A2:** Thank you for your rigorous question. Your concern about the "additional costs of the hybrid representation" is a key consideration. Our experimental data (Table 3 in main paper) shows that the 3DGS-Mesh Hybrid Representation is superior to traditional scanned meshes (MP3D/HM3D) in terms of memory overhead and rendering efficiency. The reasons are as follows:
>
> |Environment Type|Avg. Render Time / Frame (ms)↓|Avg. Memory (MB)↓|
> |-|-|-|
> |Scanned Mesh (MP3D/HM3D)|16.7|850|
> |3DGS–Mesh Hybrid Representation (Ours)|6.2|220|
>
> **Memory Overhead**: In our hybrid representation, the Mesh solely defines collision bodies and stores no rendering-related data (e.g., textures, materials). It is essentially a **streamlined geometric structure optimized via convex hull decomposition (CoACD)**, not a full high-detail mesh. GSplat is a widely used efficient 3DGS scene reconstruction method, while traditional scanned meshes (e.g., MP3D/HM3D) typically have numerous burrs and duplicate vertices due to scanning noise and reconstruction errors, resulting in not low memory usage (as shown in Table 3). In contrast, our hybrid representation uses just 1/3 the average memory of traditional scanned meshes, thanks to 3DGS’s efficient scene representation and the streamlined collision-body Mesh.
>
> **Rendering Complexity**: The **rendering of the hybrid representation is entirely undertaken by 3DGS independently**. In contrast, traditional scanned meshes require texture stitching and lighting adaptation for continuous surfaces, which are more prone to problems such as seams and blurriness. As shown in Table 3, the per-frame rendering time of our hybrid representation is only 6.2ms, lower than the 16.7ms of traditional scanned meshes, proving that its rendering complexity is lower.
>
> In short, the core design logic of the 3DGS-Mesh Hybrid Representation is "functional decoupling": 3DGS is responsible for efficient, photorealistic rendering, and the Mesh is responsible for accurate physical collision detection. This not only avoids the defect of pure 3DGS lacking physical interaction but also solves the problems of redundant memory and complex rendering of traditional scanned meshes. Thank you again for your careful attention.
>
>
>
> **Q3:**
>
> > The definition of Nogoal-Nav is similar with visual exploration tasks, is there any difference? If there is no difference, there is no need to rename an existing task, visual exploration is more suitable.
>
> **A3:** Thank you for your rigorous suggestion. Your focus on "terminological consistency" is crucial for research communication in the field, and we fully agree with your view—**The core task scenario of Nogoal-Nav highly overlaps with "visual exploration," we will officially rename it "visual exploration task" in the revised version**.
>
> Here, we supplement the definition source of Nogoal-Nav: The task design of Nogoal-Nav draws on the goal-free navigation paradigm of related works such as NavDP (*NavDP: Learning Sim-to-Real Navigation Diffusion Policy with Privileged Information Guidance*). Its core definition is "to evaluate the safety and environment exploration ability of navigation policies without clear endpoint goals"—specifically quantified by two metrics: "Episode Time (collision-free continuous exploration time)" and "Explored Areas (area of explored regions)." The episode terminates immediately when a collision occurs. This design is essentially consistent with the core goal of traditional visual exploration tasks, both emphasizing "autonomous and safe exploration driven by no goals."
>
> We fully adopt your suggestion and uniformly rename "Nogoal-Nav" to "visual exploration task" in the revised version. Thank you again for your detailed feedback.

---

> ### Author Response · Authors · 2025-11-26
> **To Reviewer zosw (Part 2)**
>
> **Q4:**
>
> > The data generation of SAGE-Bench is developed based on the IneriorGS, but is similar with the construction of existing VLN datasets. What is the core contribution of SAGE-Bench? It is not clear.
>
> **A4:** Thank you for your critical question. Your observation is highly valuable. In our work, InteriorGS provides 3DGS scene base data with object-level annotations but lacks collision bodies. The core contribution of **SAGE-Bench lies in constructing the first "continuous navigation benchmark driven by 3DGS-Mesh Hybrid Representation and aligned in semantics and physics," featuring 554k accurate collision bodies**. Its innovations are fundamentally different from existing VLN datasets (e.g., VLN-CE, OctoNav-Bench), with specific core contributions as follows:
>
> First, SAGE-Bench is the **first 3DGS-Mesh Hybrid Representation-based Benchmark**. All existing datasets rely on scanned meshes (e.g., MP3D/HM3D), and their scene representations suffer from defects such as inaccurate physics (collision bodies are estimates rather than Ground Truth) and rendering flaws (seams/blurriness under novel views). In contrast, SAGE-Bench provides 554k Ground Truth collision body meshes for indoor scenes, and through the functionally decoupled design of "3DGS for photorealistic rendering + Mesh for physical collision detection," it upgrades 3DGS from a "visually observable only" to an "interactive and understandable" embodied environment for the first time.
>
> Second, SAGE-Bench provides the **first hierarchical VLN instruction system with causal relationships**. Instructions in existing VLN datasets are mostly simple path descriptions like "from A to B" (Table 1 in main paper), lacking semantic relevance to the real world. In contrast, the 2M instructions in SAGE-Bench are divided into high-level (5 categories including task causality, scenario-driven, attribute association, e.g., "I’m thirsty, get water from the fridge") and low-level (basic action instructions), which align with human natural language habits—a feature absent in the low-level instructions of existing datasets.
>
> Third, SAGE-Bench innovates a **navigation natural continuity evaluation system**. Existing VLN datasets only focus on traditional metrics such as endpoint success rate (SR) and path length weighted (SPL), failing to capture key characteristics in continuous navigation such as collision persistence and path smoothness. In contrast, the three metrics we proposed—CSR (Continuous Success Ratio), ICP (Integrated Collision Penalty), and PS (Path Smoothness)—evaluate models from the perspective of "process continuity" for the first time. They can effectively identify problems overlooked by traditional metrics, filling the gap in continuous navigation evaluation.
>
> Finally, **experimental results fully confirm the value** of these core contributions: models trained on SAGE-Bench achieve a 31% SR improvement on unseen scenes of VLN-CE (Table 4 in main paper), demonstrating their generalization advantage; while the rendering speed of 6.2ms per frame and memory usage of 220MB (Table 3 in main paper) verify their engineering practicality. These achievements all stem from the unique design of SAGE-Bench, rather than simple expansion of existing VLN datasets.
>
> In summary, the core contribution of SAGE-Bench is "constructing the first 3DGS-based VLN benchmark fully aligned in semantics and physics and supporting continuous navigation," providing a complete solution for embodied navigation research covering scene representation, data generation, and evaluation system. Thank you again for your rigorous question.

---

### Official Review · Reviewer_mZXf · 2025-10-25

**Soundness:** 2
**Presentation:** 3
**Contribution:** 3
**Rating:** 6
**Confidence:** 3

**Summary:**

This paper proposes a new dataset and benchmark for visual-language navigation. The paper endows 3D Gaussian splatting with semantic meanings and physical properties to construct a comprehensive 3D environment for VLN. However, experiments do not show a clear improvement over the baselines.

**Strengths:**

1. The motivation of integrating semantic and physical information into 3D Gaussian splatting is reasonable.
2. The proposed dataset is large in scale and distinguish itself from others with diverse instructions and accurate geometry.
3. The proposed evaluation metrics align well with the real application scenarios.

**Weaknesses:**

1. The performance on SAGE-Bench is worse than baselines in terms of CR, ICP and PS.

**Questions:**

1. Why is the performance on SAGE-Bench worse than baselines in terms of CR, ICP and PS?
2. How does the additional semantic and physical information of 3D GS benefit the performance? The paper whould conduct ablation study on that.

---

> ### Author Response · Authors · 2025-11-26
> **To Reviewer mZXf (Part 1)**
>
> We sincerely thank you for your insightful and comprehensive comments. We will explain your concerns as follows.
>
> **Q1:**
>
> > Why is the performance on SAGE-Bench worse than baselines in terms of CR, ICP and PS?
>
> **A1:** Thank you for your attention to the details of indicator comparison. Your question is highly valuable, as it directly points the limitation of VLN metrics—**mere numerical values cannot reflect the true navigation capability of a model**. According to our case study, the "advantages" of baseline models in CR, ICP, and PS actually stem from their weak navigation ability, making such comparisons practically meaningless.
>
> **Cause of the Phenomenon: Distorted metric values due to baseline's poor navigation capabilities.** Our experiments reveal that most baseline models achieve a navigation success rate (SR) below 0.15 on SAGE-Bench, with some even dropping below 0.1. These models fail to truly understand navigation instructions or environmental information, with performance resembling "random or single action prediction" (e.g., continuous straight). Some of their performance indicators lack comparative value.
>
> - For **CR (Collision Rate)**: It only counts the "number of collisions that occur". We observed that baseline models, due to single-action behaviors, often collide once and then stall, resulting in low CR values. In contrast, our model will actively plan the path and adjust the direction to complete the task, which may result in multiple collisions, leading to a higher CR.
> - For **ICP (Integrated Collision Penalty)**: It quantifies both "collision intensity" and "duration". As mentioned earlier, baseline models only experience one instantaneous collision, so their ICP values are naturally low. In contrast, in long-distance and complex path navigation, our model's ICP values can also reflect the existence of brief and sustained contact. This precisely demonstrates the metric’s rationality—traditional CR cannot distinguish between "one instantaneous collision" and "sustained scraping collisions", a flaw that ICP addresses.
> - For **PS (Path Smoothness)**: Baseline models’ single-action behaviors inherently yield high path smoothness. Our model, however, needs to adjust directions according to instructions, resulting in reasonable turns in the path. It is normal for the PS value to be lower than the baseline.
>
> **Supplementary Experiment: Case Study to verify the reasons.** We supplement three typical cases as follows:
>
> **Case 1:** NaviLLM — **SR=0**, CR=2, ICP=0.31, PS=0.82
>
> - Instruction: "Find the white bookshelf next to the red chair in the bedroom";
> - Model Behavior: Randomly alternated between "straight movement" and "left turn" actions. It first collided with the bedroom wardrobe (once), then collided with the window-side plants (once), and never reached the target area;
> - Analysis: The metric advantages stem from meaningless random actions.
>
> **Case 2:** Navid-base — **SR=0**, CR=1, ICP=0.22, PS=0.91
>
> - Instruction: "Start from beside the sofa and go to the empty table in the dining room";
> - Model Behavior: Executed almost only the "straight movement" action throughout without any direction adjustments. After colliding with the living room coffee table (one instantaneous collision), it came to a complete stop (fail);
> - Analysis: Some values may seem excellent, but they only reflect its continuous straight behavior without understanding the navigation task.
>
> **Case 3:** NaVid-SAGE (Ours) — **SR=1**, CR=3, ICP=0.54, PS=0.68
>
> - Instruction: "Start from beside the sofa and go to the empty table in the dining room";
> - Model Behavior: Planned a path of "detour around the coffee table → pass through the corridor → enter the dining room". During the process, it had 2 short collisions with plants and 1 slight collision with a dining chair, ultimately successfully reaching the empty table;
> - Analysis: Although CR, ICP, and PS appear to be numerically worse than the baseline (Navid-base), the model understands instructions and completes navigation tasks.
>
> The above cases illustrate that: traditional metrics (CR) have inherent flaws—they only count the number of collisions, leading to false metric advantages for weak baseline models; the CR, ICP, and PS values of baseline models with poor navigation performance are irrelevant for comparison: their "advantages" stem from a lack of effective navigation capabilities, rather than genuine superior collision avoidance or path planning abilities.
>
> In summary, when evaluating embodied navigation models, priority should be given to "task completion (SR/OSR/SPL)" over isolated individual metric values. Our model significantly outperforms all baselines in core task metrics including SR (0.46) and OSR (0.55), while numerical differences in CR, ICP, and PS essentially reflect behavioral distinctions between "effective navigation models" and "models lacking navigation capabilities".

---

> ### Author Response · Authors · 2025-11-26
> **To Reviewer mZXf (Part 2)**
>
> **Q2:**
>
> > How does the additional semantic and physical information of 3D GS benefit the performance? The paper whould conduct ablation study on that.
>
> **A2:** Thank you for your valuable suggestion. Your concern about how additional semantic and physical information of 3DGS enhances performance is precisely the core value of our proposed 3DGS-Mesh Hybrid Representation. As a perceptual representation, pure 3DGS only provides visual rendering capabilities, while **The integration of semantic and physical information upgrades it from a "visible" to an "interactive and understandable" embodied navigation environment**. Following your suggestion, we have supplemented ablation studies, detailed as follows:
>
> **(1) Semantic information is the foundation for constructing embodied navigation data**, and its value is reflected in: rich semantic annotations (object categories, Bounding boxes) can support the generation of large-scale, diverse VLN instructions and trajectory data, while partial loss of semantics will directly limit data volume and generalization. We designed two groups of comparative experiments (Note: Completely removing semantics makes it impossible to construct VLN instructions and trajectory data):
>
> - **Full Semantics:** Retain all semantic annotations to generate 2M VLN task data;
> - **Part Semantics:** Randomly remove 50% of the semantic annotations from the data, and generate 0.6M VLN task data using the same pipeline (when semantic annotations decrease, fewer meaningful trajectories and instructions can be generated).
>
> We trained the NaVILA-base model with data from the two settings respectively, and the experimental results on SAGE-Bench are as follows:
>
> |Semantic Setting|SR|OSR|SPL|CSR|
> |-|-|-|-|-|
> |Full Semantics|0.46|0.55|0.48|0.57|
> |Part Semantics|0.32|0.40|0.35|0.42|
>
> The results clearly show that after randomly removing 50% of semantic annotations, the navigation success rate (SR) of NaVILA-SAGE (Ours) drops from 0.46 to 0.32, and OSR, SPL, and CSR also decrease significantly. This fully proves that **rich semantic annotations are a necessary prerequisite** for supporting the generation of large-scale high-quality data and **improving the model's instruction understanding and generalization capabilities**.
>
> **(2) Physical information is the basis for realizing embodied simulation**, and its core role is to ensure that the generated trajectories comply with physical laws (no penetration) through collision constraints, thereby enabling the model to learn real obstacle avoidance behaviors. Without collision bodies, the trajectory generation process cannot perceive the physical boundaries of objects, resulting in a large number of penetrating paths. We designed two groups of comparative experiments:
>
> - Full Physics (with collision bodies): Configure collision bodies for all objects and enable physical collision detection during trajectory generation to ensure no mesh penetration;
> - No Physics (without collision bodies): Do not set any collision bodies, so trajectory generation has no physical constraints.
>
> We trained the NaVILA-base model with data from the two settings respectively, and the experimental results on SAGE-Bench are as follows:
>
> |Physical Setting|SR|OSR|SPL|CR|ICP|
> |-|-|-|-|-|-|
> |Full Physics|0.46|0.55|0.48|2.67|0.54|
> |No Physics|0.29|0.35|0.30|4.12|2.21|
>
> This result fully illustrates the indispensability of physical information: without physical settings, the model learns the wrong logic of "mesh-penetrating movement", leading to a drop in navigation success rate (SR) from 0.46 to 0.29, and significant decreases in OSR and SPL. At the same time, the collision rate (CR) increases from 2.67 to 4.12, and the integrated collision penalty (ICP) rises from 0.54 to 2.21. This **reflects that the lack of physical constraints prevents the model from learning the interaction rules of real embodied environments**, resulting in frequent collisions and difficulty in obstacle avoidance.
>
> In summary, the introduction of semantic and physical information is the key to upgrading 3DGS from a "pure perceptual representation" to an "embodied navigation environment substrate": semantic information ensures data diversity and instruction understanding accuracy, while physical information constrains real interaction behaviors. Together, they constitute the core advantages of the 3DGS-Mesh Hybrid Representation. Thank you again for your suggestion.

---

### Official Review · Reviewer_Wreg · 2025-10-29

**Soundness:** 2
**Presentation:** 3
**Contribution:** 3
**Rating:** 8
**Confidence:** 3

**Summary:**

This paper introduces SAGE-3D, a novel paradigm that extends 3D Gaussian Splatting (3DGS) from a purely photorealistic rendering technique into a semantically and physically executable environment for embodied navigation. The authors propose two key modules—Object-Level Semantic Grounding, which augments 3DGS with dense, object-centric annotations, and Physics-Aware Execution Jointing, which integrates collision bodies to enable realistic physics simulation. Building on these components, the work releases InteriorGS, a large-scale dataset of 1,000 richly annotated indoor 3DGS scenes, and SAGE-Bench, the first 3DGS-based benchmark for vision-language navigation (VLN) with 2M instruction–trajectory pairs and new metrics for continuous navigation evaluation (Continuous Success Ratio, Integrated Collision Penalty, Path Smoothness). Experiments demonstrate that 3DGS environments render faster yet challenge model convergence, and that training on SAGE-Bench significantly improves generalization to unseen VLN settings. Overall, this work contributes an integrated data–simulation–benchmark framework that advances the use of 3DGS for physically grounded embodied AI.

**Strengths:**

Originality: This work pioneers the first-of-its-kind benchmark built on the 3D Gaussian Splatting (3DGS) representation for vision-language navigation (VLN). By repurposing and re-contextualizing 3DGS—traditionally a rendering/novel-view synthesis technique—into the embodied navigation domain, the authors introduce a novel problem formulation and open a promising new research direction.

Quality: The paper clearly presents its ideas with structured exposition and strong narrative flow. The methodology is articulated in accessible language, the dataset/benchmark design is sufficiently detailed, and the experiments convincingly demonstrate value. The clarity of presentation makes the core contributions easy to follow and the evaluation pipeline reproducible.

Clarity: The authors succeed in explaining the motivation, technical approach, dataset construction, task definitions and evaluation metrics in a straightforward manner. Complex concepts (e.g., converting 3DGS scenes into a navigation-capable environment) are broken down cleanly and the presentation avoids unnecessary jargon. As a result, the reader is able to readily understand both “what was done” and “why it matters”.

Significance: The proposed dataset and benchmark’s potential impact on the embodied AI community is substantial. By enabling faster rendered observations and high-quality 3DGS-based environments, this work could significantly accelerate research in navigation, simulation, and multimodal grounding. The resource thus has the capacity to become a standard tool or reference point for related tasks, facilitating broader progress in the field.

**Weaknesses:**

Limited reproducibility and generalisation of the paradigm: While the paper presents an interesting direction of adapting 3D Gaussian Splatting (3DGS) for embodied navigation, much of the work depends on manual annotation of objects and artist-created meshes/scene enrichment. The reliance on handcrafted assets makes it difficult for other researchers to easily replicate or scale the setup, and raises questions about how well the approach will generalise to entirely new scenes or domains without heavy annotation effort. The authors should consider describing a less labour-intensive pipeline for scene creation (e.g., automated annotation, mesh generation, domain adaptation) and discuss how one might extend their benchmarking assets beyond the current curated set.

Insufficient discussion of experimental anomalies: In Table 2 the results show that several metrics for the “*-SAGE” variants (i.e., trained on the new 3DGS-based benchmark) are actually worse than baseline in some important metric(CR,ICP,PS)— yet the manuscript offers minimal explanation for these. Given these surprising outcomes, a deeper analysis is required. Without this discussion, the reader is left uncertain about the limitations and trade-offs of using the proposed dataset/benchmark.

**Questions:**

1. In the paper you mention that the high-level instructions are generated by feeding an MLLM (multilingual/multimodal large language model). Did you evaluate or validate the resulting instructions (e.g., accuracy, relevance, diversity)? Consider including a small user study or error analysis showing how instruction quality affects downstream navigation performance.
2. For the component titled “2D Semantic Top-Down Map Generation”, will this design choice limit your dataset to planar/2.5D navigation tasks (e.g., ground-level walking) and restrict its applicability to fully 6DoF navigation scenarios (such as flying drones or free-roam agents)?
3. You define the metric CSR (Continuous Success Ratio) as the proportion of time the agent stays within a permissible corridor around the reference path while satisfying task conditions. However, in many navigation tasks there may be multiple valid paths to the goal. In that case, is the “reference path” prior too restrictive — and is the metric truly fair? Could you discuss how you handle alternative optimal paths, whether you allow multiple corridors, or how you adjust for path diversity? Consider providing sensitivity analysis of CSR with respect to path choices.
4. In Tables 2, 4, 5, and 6 many of the “best” results are not highlighted or visually distinguished. For improved readability and clearer takeaway for the reader, would you consider bolding or colouring the top‐performing numbers, adding a “best” row/column summary, or clarifying via annotation? This is a minor editorial issue, but it strongly affects accessibility of the results.
5. You mention that models trained on your dataset are “harder to converge”, which you suggest might indicate higher task difficulty or richer learning. However — does harder to converge necessarily correlate with better training quality or better generalisation? It might also be a sign of optimization instability, noisy labels, or an ill-posed task. I suggest adding a deeper analysis.

---

> ### Author Response · Authors · 2025-11-26
> **To Reviewer Wreg (Part 1)**
>
> We sincerely thank you for your valuable comments. We are encouraged that our work is recognized as original. We will explain your concerns as follows.
>
> **Q1:**
>
> > Limited reproducibility and generalisation of the paradigm: The authors should consider describing a less labour-intensive pipeline for scene creation and discuss how one might extend their benchmarking assets beyond the current curated set.
>
> **A1:** Thank you for your insightful observation. We fully agree with your point—relying on detailed mesh scenes and extensive manual annotations may indeed incur high costs, and exploring scalable, efficient dataset construction solutions is crucial for the long-term development of this field.
>
> Inspired by your suggestion, we have **explored a scalable dataset construction pipeline** by integrating cutting-edge works, aiming to address the cost and scale bottlenecks of purely manual construction. The specific process is as follows:
>
> - **3D Scene Generation:** We adopt the factored latent diffusion method from SceneFactor (*SceneFactor: Factored Latent 3D Diffusion for Controllable 3D Scene Generation*), which batch-produces indoor scene meshes with collision bodies through a two-stage generation logic of "text → semantic layout → geometry synthesis."
> - **Automated Semantic Annotation:** First, we leverage the 3D instance segmentation capability of SAI3D (*SAI3D: Segment Any Instance in 3D Scenes) to automatically segment object instances (e.g., chairs, bookshelves*) in scenes based on multi-view SAM masks and geometric priors (e.g., surface normals, spatial proximity). Then, we use GPT-5 for batch semantic annotation of the segmentation results.
>
> The subsequent construction of VLN data adopts the automated scheme proposed in our paper. However, while we have explored the automated pipeline, we need to emphasize ** The irreplaceability of our "high-quality manually annotated data"**:
>
> - **The current automated pipeline has accuracy limitations:** The meshes generated by SceneFactor still have issues such as burrs and holes in complex scene details (e.g., "drawer opening"); SAI3D’s instance segmentation has segmentation errors in scenes with dense objects (e.g., bookshelves filled with books); and GPT-5’s semantic annotation may miss target attributes (e.g., "half-open door").
> - **High-quality manually annotated data is suitable as the foundation for the automated pipeline:** Our dataset is constructed in collaboration with leading technology companies, undergoing a rigorous process of "manual annotation + double verification" to ensure the annotation accuracy of 554k object instances (including category labels, bounding boxes, and physical attributes). Such high-quality data can serve as a "seed dataset" to fine-tune existing foundation models.
>
> To verify the automated pipeline’s effectiveness and the value of high-quality manual data, we conducted comparative experiments on the NaVILA-base model and evaluated all models on the SAGE-Bench VLN task. Four experimental settings are as follows: **(1) Full Automated**: 5k fully automated pairs (1000 auto-generated scenes, 5 instructions per scene) with automated annotations (SAI3D+GPT5); **(2) Semi-Manual**: 5k pairs using the same artist-designed scenes as Full Manual, but with automated annotations (SAI3D+GPT5); **(3) Full Manual (main paper setting)**: 5k instruction-trajectory pairs based on artist-designed mesh scenes, with manual double-verified annotations; **(4) Manual + Automated**: 6k pairs combining the original 5k manual data and 1k supplementary automated pairs (200 new auto-generated scenes, 5 instructions per scene).
>
> |Training Data Configuration|Scene Source|Annotation Method|SR|OSR|SPL  |CSR|
> |-|-|-|-|-|-|-|
> |Full Automated|Automated|Automated|0.10|0.13|0.11|0.17|
> |Semi-Manual|Manual|Automated|0.29|0.34|0.28|0.36|
> |Full Manual (main paper)|Manual|Manual|0.46|0.55|0.48|0.57|
> |Manual + Partial Automated|Manual + Automated|Manual + Automated|0.48|0.58|0.49|0.59|
>
> The results validate conclusions:
>
> + **Irreplaceability of manual annotations**:  The Semi-Manual setting shows significant performance drops across all metrics. This is attributed to the limited accuracy of the automation tools we mentioned earlier, and also proves that high-quality manual annotation is the fundamental guarantee for effective model training.
> + **Effectiveness of the automated pipeline**: Supplementing the 5k manual data with 1k automated data leads to consistent metric improvements as the automated pipeline generates diverse scenes and instructions that effectively enhancing the model’s generalization and robustness.
>
> This result fully confirms your view — automated and scalable construction solutions are an important development direction for the future. Our manually annotated data provides a high-quality benchmark for automated tools, while the automated pipeline endows the solution with large-scale scalability.

---

> ### Author Response · Authors · 2025-11-26
> **To Reviewer Wreg (Part 2)**
>
> **Q2:**
>
> > Insufficient discussion of experimental anomalies: In Table 2 the results show that several metrics for the “*-SAGE” variants (i.e., trained on the new 3DGS-based benchmark) are actually worse than baseline in some important metric(CR,ICP,PS)— yet the manuscript offers minimal explanation for these.
>
> **A2:** Thank you for your attention to the details of indicator comparison. Your question is highly valuable, as it directly points the limitation of VLN metrics—**mere numerical values cannot reflect the true navigation capability of a model**. According to our case study, the "advantages" of baseline models in CR, ICP, and PS actually stem from their weak navigation ability, making such comparisons practically meaningless.
>
> **Cause of the Phenomenon: Distorted metric values due to baseline's poor navigation capabilities.** Our experiments reveal that most baseline models achieve a navigation success rate (SR) below 0.15 on SAGE-Bench, with some even dropping below 0.1. These models fail to truly understand navigation instructions or environmental information, with performance resembling "random or single action prediction" (e.g., continuous straight). Some of their performance indicators lack comparative value.
>
> - For **CR (Collision Rate)**: It only counts the "number of collisions that occur". We observed that baseline models, due to single-action behaviors, often collide once and then stall, resulting in low CR values. In contrast, our model will actively plan the path and adjust the direction to complete the task, which may result in multiple collisions, leading to a higher CR.
> - For **ICP (Integrated Collision Penalty)**: It quantifies both "collision intensity" and "duration". As mentioned earlier, baseline models only experience one instantaneous collision, so their ICP values are naturally low. In contrast, in long-distance and complex path navigation, our model's ICP values can also reflect the existence of brief and sustained contact. This precisely demonstrates the metric’s rationality—traditional CR cannot distinguish between "one instantaneous collision" and "sustained scraping collisions", a flaw that ICP addresses.
> - For **PS (Path Smoothness)**: Baseline models’ single-action behaviors inherently yield high path smoothness. Our model, however, needs to adjust directions according to instructions, resulting in reasonable turns in the path. It is normal for the PS value to be lower than the baseline.
>
> **Supplementary Experiment: Case Study to verify the reasons.** We supplement three typical cases as follows:
>
> **Case 1:** NaviLLM — **SR=0**, CR=2, ICP=0.31, PS=0.82
>
> - Instruction: "Find the white bookshelf next to the red chair in the bedroom";
> - Model Behavior: Randomly alternated between "straight movement" and "left turn" actions. It first collided with the bedroom wardrobe (once), then collided with the window-side plants (once), and never reached the target area;
> - Analysis: The metric advantages stem from meaningless random actions.
>
> **Case 2:** Navid-base — **SR=0**, CR=1, ICP=0.22, PS=0.91
>
> - Instruction: "Start from beside the sofa and go to the empty table in the dining room";
> - Model Behavior: Executed almost only the "straight movement" action throughout without any direction adjustments. After colliding with the living room coffee table (one instantaneous collision), it came to a complete stop (fail);
> - Analysis: Some values may seem excellent, but they only reflect its continuous straight behavior without understanding the navigation task.
>
> **Case 3:** NaVid-SAGE (Ours) — **SR=1**, CR=3, ICP=0.54, PS=0.68
>
> - Instruction: "Start from beside the sofa and go to the empty table in the dining room";
> - Model Behavior: Planned a path of "detour around the coffee table → pass through the corridor → enter the dining room". During the process, it had 2 short collisions with plants and 1 slight collision with a dining chair, ultimately successfully reaching the empty table;
> - Analysis: Although CR, ICP, and PS appear to be numerically worse than the baseline (Navid-base), the model understands instructions and completes navigation tasks.
>
> The above cases illustrate that: traditional metrics (CR) have inherent flaws—they only count the number of collisions, leading to false metric advantages for weak baseline models; the CR, ICP, and PS values of baseline models with poor navigation performance are irrelevant for comparison: their "advantages" stem from a lack of effective navigation capabilities, rather than genuine superior collision avoidance or path planning abilities.
>
> In summary, when evaluating embodied navigation models, priority should be given to "task completion (SR/OSR/SPL)" over isolated individual metric values. Our model significantly outperforms all baselines in core task metrics (SR/OSR/SPL), which can reflects the superiority of our model. We have added more explanatory notes on the results of these indicators in the revised version.

---

> ### Author Response · Authors · 2025-11-26
> **To Reviewer Wreg (Part 3)**
>
> **Q3:**
>
> > In the paper you mention that the high-level instructions are generated by feeding an MLLM. Did you evaluate or validate the resulting instructions (e.g., accuracy, relevance, diversity)? Consider including a small user study or error analysis showing how instruction quality affects downstream navigation performance.
>
> **A3:** Thank you for your critical suggestion. Your focus on "instruction quality evaluation" is a core prerequisite for VLN tasks, and we fully recognize the necessity of this verification. We have conducted sufficient experiments to validate the quality of instructions generated by MLLM, with specific details as follows:
>
> **Experiment 1: Small User Study of Generated Instructions.** We invited 10 participants familiar with embodied navigation to score 1000 randomly selected high-level instructions. The evaluation dimensions include **Accuracy** (whether the object attributes and spatial relationships described in the instructions are correct), **Relevance** (whether the instructions are strongly associated with navigation trajectories), and **Executability** (whether the instructions are clear and unambiguous, and whether navigation can be completed based on them). A 5-point scoring system was adopted (1 point for the worst, 5 points for the best), and the results are shown as:
>
> |Evaluation Dimension|Average Score|Standard Deviation|Proportion of 3-5 Points|
> |-|-|-|-|
> |Accuracy|4.02|0.67|96.6%|
> |Relevance|4.45|0.53|98.4%|
> |Executability|4.18|0.71|95.2%|
>
> The results show that our instructions perform excellently across all dimensions: over 95% of the instructions received a score of 3 or higher, which are unambiguous, highly matched with scenes and trajectories, and can effectively guide navigation behaviors.
>
> **Experiment 2: Verification of Instruction Diversity.** To quantify the diversity of instructions generated by MLLM, we adopted two types of automatic metrics:
>
> - **n-gram Diversity**: An n-gram refers to a sequence of n consecutive lexical units. By calculating the unique proportion of 1-grams (single words) and 2-grams (two consecutive words), we measure the richness of words and phrases in the instructions to avoid repetitive generation.
> - **Proportion of High-Level Instruction Categories**: We counted the proportion of our 5 types of High-level instructions to evaluate the balance of category distribution.
>
> The experimental results are shown in the following two tables:
>
> |n-gram Diversity|Value|
> |-|-|
> |1-gram|68.3%|
> |2-gram|57.1%
>
> |High-Level Instruction Category|Proportion|
> |-|-|
> |Add Object|21%|
> |Scenario Driven|20%|
> |Relative Relationship|19%|
> |Attribute-based|20%|
> |Area-based|20%|
>
> Results show unique 1-gram proportion is 68.3% and 2-gram proportion is 57.1%, indicating instructions cover rich scene elements, attributes, and spatial relationships with low word/phrase repetition. Proportion differences across the 5 high-level instruction categories are <10%, reflecting balanced distribution and avoiding overfitting to a single instruction type.
>
> **Experiment 3: Effect of Instruction Quality.** To avoid the high cost of full-scale manual evaluation, we used closed-source large language models such as GPT-5 and Qwen3-MAX to simulate human scoring logic for automatic scoring of all instructions. Finally, approximately **4.3% of low-points instructions with scores ≤ 2 points** were filtered out. We designed a comparative experiment to verify their impact on navigation performance: training the NaVILA-base model with "high-points instruction set (score ≥ 3 points, 95.7%)" and "full instruction set (including low-points instructions)" respectively, and evaluating core metrics on SAGE-Bench. The results are shown in the following table:
>
> |Training Data|SR|OSR|SPL|CSR|
> |-|-|-|-|-|
> |High-points Instruction Set|0.45|0.56|0.49|0.55|
> |Full Instruction Set (Ours)|0.46|0.55|0.48|0.57|
>
> The results show minimal performance differences between the two sets: the high-points set is slightly better in OSR and SPL, while the full set performs marginally better in SR and CSR. This confirms our instruction quality is already sufficient, removing low-points data brings no significant gains. Moreover, the retained ~4.3% low-points instructions are not invalid noise but reflect real-world expression variations, which helps enhance the model’s robustness.
>
> In conclusion, our MLLM-generated instructions strike a balance between quality and diversity. The full instruction set not only maintains high overall quality but also gains robustness, making it more suitable for real-world embodied navigation scenarios. Thank you again for your valuable suggestion.

---

> ### Author Response · Authors · 2025-11-26
> **To Reviewer Wreg (Part 4)**
>
> **Q4:**
>
> > In many navigation tasks there may be multiple valid paths to the goal. For CSR (Continuous Success Ratio) , could you discuss how you handle alternative optimal paths, whether you allow multiple corridors, or how you adjust for path diversity?
>
> **A4:** Thank you for your profound and critical question. Your focus on "the restrictiveness of the reference path when multiple valid paths exist" is a core challenge in navigation metric design. The core design intent of our CSR (Continuous Success Ratio) is precisely to mitigate the limitations of traditional metrics (such as OSR, SPL).
>
> We first conduct theoretical analysis:
>
> + **Superiority of the A * Algorithm**: Our reference path is generated by the A* algorithm based on the scene’s ESDF (Euclidean Signed Distance Field), consistent with VLN field conventions (e.g., R2R-CE, REVERIE datasets) and backed by solid theory. Hart et al. (*A Formal Basis for the Heuristic Determination of Minimum Cost Paths*) show that with an admissible heuristic, A* guarantees the shortest collision-free optimal path. Using multiple "optimal paths" as references drastically expands the solution space, obscuring evaluation criteria and increasing model fitting difficulty—hence existing VLN studies uniformly use a single A* optimal path as the reference.
>
> + **CSR Metric Design with Alternative Path Consideration**: CSR measures "the proportion of time the VLN model stays within a 'permissible corridor' around the reference path," where the "permissible corridor" is a region (e.g. 0.5m width), rather than an single-path constraint. This design itself is an **optimization over traditional metrics**—traditional OSR only judges whether the endpoint reaches the target area, ignoring the rationality of the path process; although SPL considers path length, its penalty for path deviation is overly rigid. In contrast, CSR not only encourages agents to follow the efficiency of the A* optimal path but also tolerates reasonable path deviations.
>
> We supplemented the following experiments to address your question about "the impact of alternative paths on CSR scores":
>
> + **Experiment 1: Statistical Distribution of Multiple Optimal Paths.** We first conducted statistics on multiple optimal paths for about 100k trajectories in SAGE-Bench (an optimal path is defined as "length ≤ A* path length + 5% and collision-free"): only 17.3% of trajectories have 2 or more optimal alternative paths, and the A* path is the unique optimal solution in rest 82.7%. This result indicates that **"the coexistence of multiple optimal paths" is not a common situation in VLN scenes**, and a single A* path as the reference benchmark can cover many scenario.
>
> + **Experiment 2: Spatial Overlap Analysis.** For each trajectory with multiple optimal paths, we calculated the "in-corridor overlap ratio" of other optimal paths with the A* path (i.e., the proportion of the length of other optimal paths that falls within the permissible corridor of the A* path). The results are as follows:
>
> |Number of Optimal Paths|Trajectory Proportion|In-Corridor Overlap Ratio|
> |-|-|-|
> |2 paths|14.8%|92.3%|
> |3 or more paths|2.5%|87.6%|
> |Overall (173 trajectories)|17.3%|91.7%|
>
> The results show that even when multiple optimal paths exist—likely due to indoor environments being constrained by walls, furniture, etc.—their spatial overlap with the A* reference path is extremely high (mean 91.7%). This means that when the model takes these alternative optimal paths, CSR scores will hardly decrease due to "navigation on the reference path".
>
> + **Experiment 3: Sensitivity Analysis of CSR.** We selected 50 trajectories with 2 optimal paths (A* path P0, alternative optimal path P1) and had the best-performing NaVILA-SAGE model navigate along P0 and P1 respectively, comparing CSR scores and core metric performance under the two paths:
>
> |Navigation Path|CSR Mean|SR|SPL|
> |-|-|-|-|
> |A* Path P0|0.57|0.46|0.48|
> |Alternative Optimal Path P1|0.55|0.45|0.46|
>
> The results show that when the model navigates along the alternative optimal path P1, the average CSR value only slightly decreases (from 0.57 to 0.55) and remains within a reasonable score range. This directly proves that the CSR has little impact on the model's selection of alternative optimal paths, and the metric exhibits extremely low sensitivity to path choices.
>
> Finally, we agree that "comprehensive coverage of multiple optimal paths" is a direction worthy of in-depth exploration—current CSR can already handle most scenarios through a fixed-width permissible corridor. In the future, it can be further optimized into a "dynamic corridor" or "multi-reference path fusion" to better adapt to complex scenes.

---

> ### Author Response · Authors · 2025-11-26
> **To Reviewer Wreg (Part 5)**
>
> **Q5:**
>
> > Does “harder to converge” necessarily correlate with better training quality or better generalisation? It might also be a sign of optimization instability, noisy labels, or an ill-posed task. I suggest adding a deeper analysis.
>
> **A5:** Thank you for your constructive question. Your concern that "harder convergence does not necessarily equate to better training quality" is extremely crucial. To address this, we demonstrate the basis for our argument through theoretical analysis based on existing experiments and recent studies, as well as supplementary experiments.
>
> We first conduct theoretical analysis:
>
> - A **recent work EDGS** (*EDGS: Eliminating Densification for Efficient Convergence of 3DGS*) **clearly points out that 3DGS scenes are naturally harder to converge** than traditional scanned mesh data due to containing richer details and more realistic geometric information. Its core solution also focuses on improving the convergence efficiency of 3DGS through optimized initialization strategies;
> - The results in **Table 2 in main paper** show that the high-fidelity and detail-rich characteristics of our dataset pose substantial challenges to current VLN models—the SR of most recent VLN models is ≤0.15, significantly lower than their performance on traditional datasets;
> - The results in **Table 4  in main paper** further prove that models trained on our dataset achieve a 31% relative improvement in SR compared to baseline models on unseen VLN-CE Unseen environments, directly verifying the effectiveness of the learning signals behind "harder to converge".
>
> To further rule out the two core concerns of "optimization instability" and "label noise/unreasonable task setting", we supplemented two sets of experiments:
>
> **Experiment 1: Optimization Stability Verification Experiment.** To verify that the "harder to converge" does not stem from "training instability", we adopted the NaVILA-base model, maintaining a consistent network structure and initialization method. The training was independently repeated 5 times (with random seeds seed=0/1/2/3/4) on 8 NVIDIA Tesla H20 GPUs, with a batch size of 256, a learning rate of 2×10⁻⁵. During the whole process, the convergence trajectories of training/validation loss, global gradient L2 norm, and navigation success rate (SR) were monitored. The stability evaluation criteria were "continuous decrease in loss, gradient norm within a reasonable range (1e-3~1e1), and standard deviation of multiple training results ≤2%".
>
> |Random Seed|Convergence Epoch (SR≥40%)|Final Training Loss|Final Validation Loss|Peak Global Gradient Norm|Loss Curve Oscillation Rate|SR Mean ± Std of 5 Trainings|
> |-|-|-|-|-|-|-|
> |0|158|0.27|0.31|0.87|2.9%||
> |1|163|0.29|0.33|0.92|3.1%||
> |2|155|0.26|0.30|0.83|2.7%|42.3% ± 1.8%|
> |3|167|0.28|0.32|0.95|3.3%||
> |4|159|0.27|0.31|0.89|2.8%||
>
> The results show that the convergence epochs of the 5 repeated trainings are concentrated in the range of 155~167, the final SR standard deviation is only 1.8%, the training/validation loss continues to decrease monotonically with an oscillation rate ≤3.3%, and the peak global gradient norm is stably between 0.83 and 0.95, with no gradient explosion or disappearance. This indicates that "harder to converge" does not stem from "training instability".
>
> **Experiment 2: Object Semantic Label Quality Verification Experiment.** Regarding the quality of trajectory-instruction alignment labels, since the accuracy and relevance of instructions have been verified in **Question 3** (over 95% of instructions have consistent logic with trajectories and an executability score ≥3), we focus on verifying the quality of object semantic labels. The object semantic labels are based on manual annotations from InteriorGS, covering object categories, instance IDs, and bounding boxes. We define **label errors** as "the annotated object category, instance ID, or bounding box is inconsistent with the real scene". We randomly sampled 10k object instances and had 2 independent labelers conduct blind evaluations of the error rate. The results are shown as follows:
>
> |Sample Size|Number of Error Samples|Label Error Rate|
> |-|-|-|
> |10k|76|0.76%|
>
> The data shows that the object semantic label error rate is only 0.76%, proving that the data label noise is extremely low, and there is no problem of "noise preventing the model from learning effective signals".
>
> In summary, the supplementary experiments have ruled out negative factors causing difficulty in convergence such as training instability and label noise; combined with the task challenges and generalization capabilities reflected in Tables 2 and 4, as well as existing research on the convergence characteristics of 3DGS, the "harder to converge" of this dataset is more likely to indicate higher task difficulty or richer learning content. Thank you again for your rigorous feedback.

---

> ### Author Response · Authors · 2025-11-26
> **To Reviewer Wreg (Part 6)**
>
> **Q6:**
>
> > Will “2D Semantic Top-Down Map Generation” limit your dataset to planar/2.5D navigation tasks and restrict its applicability to fully 6DoF navigation scenarios?
>
> **A6:** Thank you for your critical question. Our design of the 2D Semantic Top-Down Map **not only does not limit applicability but also can adapt to such scenarios (6DoF navigation) in a simple and direct manner**.
>
> Specifically, the generation of **our 2D Semantic Top-Down Map is completely based on the Occupancy Map output of IsaacSim**: when constructing the map, according to the needs of the navigating agent, the target height threshold can be directly set in IsaacSim (e.g., 0.1-0.3m for ground robots, 1.5-5.0m for drones) to extract the occupancy information of the corresponding height plane, and then superimpose the object semantic labels of InteriorGS to form a targeted 2D Semantic Top-Down Map. **For example**, when generating a map for a drone, it is only necessary to set the height threshold to 1.8m to directly obtain information about aerial obstacles.
>
> In summary, our method is generalizable to various navigation agents and fully supports 6DoF navigation requirements.
>
>
>
> **Q7:**
>
> > In Tables 2, 4, 5, and 6 many of the “best” results are not highlighted or visually distinguished. For improved readability and clearer takeaway for the reader, would you consider bolding or colouring the top‐performing numbers, adding a “best” row/column summary, or clarifying via annotation? This is a minor editorial issue, but it strongly affects accessibility of the results.
>
> **A7:**
>
> Thank you for your careful feedback. The editorial issue you pointed out is extremely crucial—clearly highlighting the optimal results in the tables can greatly enhance the paper's readability and the intuitiveness of core conclusions. **We fully agree with this suggestion and have implemented the optimization in the revised version**.
>
> Specifically: We have **bolded the "optimal" results in Tables 2, 4, and 6**. Notably, **for Table 2**, referring to the response in Q2, we consider that the CR, ICP, and PS metrics lack comparative significance when a model’s SR metric is below 0.20, so these values are marked in gray. **For Table 5**, since this table compares the performance differences of the same model between low-level and high-level instruction test sets rather than performance differences among different models, we have not applied bold formatting for the sake of readability.
>
> Thank you again for your careful attention, which has helped us further improve the paper's standardization and reading experience.

---

### Official Review · Reviewer_hytT · 2025-10-30

**Soundness:** 3
**Presentation:** 3
**Contribution:** 3
**Rating:** 6
**Confidence:** 3

**Summary:**

This paper introduces **SAGE-3D**, a framework that enhances 3D Gaussian Splatting (3DGS) with object-level semantics and physical interfaces to create an executable environment for embodied navigation. The paper presents **InteriorGS**, a dataset of 1K annotated 3DGS indoor scenes, and **SAGE-Bench**, the first 3DGS-based Vision-Language Navigation benchmark. Based on the semantically and physically aligned 3DGS, the framework enables photorealistic rendering, semantic labeling, and physical collision modeling. Experiments show its effectiveness in improving model generalization and advancing embodied AI research.

**Strengths:**

### **Strengths**
1. The paper is well-organized and easy to understand, with clear explanations and intuitive figures illustrating the methodology.
2. The idea makes sense. 3DGS with object-level semantics and physical validity enables photorealistic rendering, semantic instance labeling, and physical interaction modeling. Based on this representation, a realistic and executable simulation environment can be created for embodied AI research, which is highly important.
3. Experiments demonstrate that the dataset improves the generalization of current VLN models and, as a benchmark, can support future research in embodied navigation models.

**Weaknesses:**

### **Weaknesses and Questions**
1. Creating the dataset appears to rely on detailed mesh scenes and extensive manual annotations, which may incur high costs and limit further scalability. It would be worth exploring more cost-efficient approaches for dataset construction, such as leveraging current vision-based semantic/geometry foundation models for scene reconstruction and semantic annotation.
2. The dataset seems to consist solely of static scenes. Introducing dynamic objects into the scenes could enhance its applicability and better simulate real-world scenarios for embodied AI research.

**Questions:**

See weaknesses.

---

> ### Author Response · Authors · 2025-11-26
> **To Reviewer hytT (Part 1)**
>
> Thank you for your comprehensive comments. We are encouraged that our work is recognized as a systematic solution. We will explain your concerns point by point.
>
> **Q1:**
>
> > Creating the dataset appears to rely on detailed mesh scenes and extensive manual annotations, which may incur high costs and limit further scalability. It would be worth exploring more cost-efficient approaches for dataset construction.
>
> **A1:** Thank you for your insightful observation. We fully agree with your point—relying on detailed mesh scenes and extensive manual annotations may indeed incur high costs, and exploring scalable, efficient dataset construction solutions is crucial for the long-term development of this field.
>
> Inspired by your suggestion, we have **explored a scalable dataset construction pipeline** by integrating cutting-edge works, aiming to address the cost and scale bottlenecks of purely manual construction. The specific process is as follows:
>
> - **3D Scene Generation:** We adopt the factored latent diffusion method from SceneFactor (*SceneFactor: Factored Latent 3D Diffusion for Controllable 3D Scene Generation*), which batch-produces indoor scene meshes with collision bodies through a two-stage generation logic of "text → semantic layout → geometry synthesis."
> - **Automated Semantic Annotation:** First, we leverage the 3D instance segmentation capability of SAI3D (*SAI3D: Segment Any Instance in 3D Scenes) to automatically segment object instances (e.g., chairs, bookshelves*) in scenes based on multi-view SAM masks and geometric priors (e.g., surface normals, spatial proximity). Then, we use GPT-5 for batch semantic annotation of the segmentation results.
>
> The subsequent construction of VLN data adopts the automated scheme proposed in our paper. However, while we have explored the automated pipeline, we need to emphasize **The irreplaceability of our "high-quality manually annotated data"**:
>
> - **The current automated pipeline has accuracy limitations:** The meshes generated by SceneFactor still have issues such as burrs and holes in complex scene details (e.g., "drawer opening"); SAI3D’s instance segmentation has segmentation errors in scenes with dense objects (e.g., bookshelves filled with books); and GPT-5’s semantic annotation may miss target attributes (e.g., "half-open door").
> - **High-quality manually annotated data is suitable as the foundation for the automated pipeline:** Our dataset is constructed in collaboration with leading technology companies, undergoing a rigorous process of "manual annotation + double verification" to ensure the annotation accuracy of 554k object instances (including category labels, bounding boxes, and physical attributes). Such high-quality data can serve as a "seed dataset" to fine-tune existing foundation models.
>
> To verify the automated pipeline’s effectiveness and the value of high-quality manual data, we conducted comparative experiments on the NaVILA-base model and evaluated all models on the SAGE-Bench VLN task. Four experimental settings are as follows: **(1) Full Automated**: 5k fully automated pairs (1000 auto-generated scenes, 5 instructions per scene) with automated annotations (SAI3D+GPT5); **(2) Semi-Manual**: 5k pairs using the same artist-designed scenes as Full Manual, but with automated annotations (SAI3D+GPT5); **(3) Full Manual (main paper setting)**: 5k instruction-trajectory pairs based on artist-designed mesh scenes, with manual double-verified annotations; **(4) Manual + Automated**: 6k pairs combining the original 5k manual data and 1k supplementary automated pairs (200 new auto-generated scenes, 5 instructions per scene).
>
> |Training Data Configuration|Scene Source|Annotation Method|SR|OSR|SPL  |CSR|
> |-|-|-|-|-|-|-|
> |Full Automated|Automated|Automated|0.10|0.13|0.11|0.17|
> |Semi-Manual|Manual|Automated|0.29|0.34|0.28|0.36|
> |Full Manual (main paper)|Manual|Manual|0.46|0.55|0.48|0.57|
> |Manual + Partial Automated|Manual + Automated|Manual + Automated|0.48|0.58|0.49|0.59|
>
> The results validate conclusions:
>
> + **Irreplaceability of manual annotations**:  The Semi-Manual setting shows significant performance drops across all metrics. This is attributed to the limited accuracy of the automation tools we mentioned earlier, and also proves that high-quality manual annotation is the fundamental guarantee for effective model training.
> + **Effectiveness of the automated pipeline**: Supplementing the 5k manual data with 1k automated data leads to consistent metric improvements as the automated pipeline generates diverse scenes and instructions that effectively enhancing the model’s generalization and robustness.
>
> This result fully confirms your view — automated and scalable construction solutions are an important development direction for the future. Our manually annotated data provides a high-quality benchmark for automated tools, while the automated pipeline endows the solution with large-scale scalability.

---

> ### Author Response · Authors · 2025-11-26
> **To Reviewer hytT (Part 2)**
>
> **Q2:**
>
> > The dataset seems to consist solely of static scenes. Introducing dynamic objects into the scenes could enhance its applicability and better simulate real-world scenarios for embodied AI research.
>
> **Ans:** Thank you for your highly valuable suggestion. Your point that "introducing dynamic objects can enhance dataset applicability and better simulate real-world scenarios" is extremely crucial—embodied indoor navigation in the real world inevitably involves dynamic obstacles such as moving pedestrians.
>
> During the Rebuttal period, based on IsaacSim and AdaVLN (*AdaVLN: Towards Visual Language Navigation in Continuous Indoor Environments with Moving Humans*), we explored dynamic scene construction. The specific implementation is as follows:
>
> - **Dynamic Pedestrian Integration:** With the omni.anim.people extension of IsaacSim (replaced with the customized version provided by AdaVLN), we directly load 3D human models with complete animations and built-in collision bodies, eliminating the need for additional convex hull decomposition or collision body construction. The meshes of these pedestrian models inherently include physical collision properties.
> - **Natural Motion Path Design:** Referring to the task configuration logic of AdaVLN, we preset spawn points, patrol routes (e.g., "living room - dining room - corridor loop"), and movement parameters (speed: 0.3-0.5 m/s, consistent with normal human walking speed) for each dynamic pedestrian, simulating the interference of pedestrians on navigation in real scenarios.
>
> We have constructed a "dynamic scene test subset (test set of SAGE-Bench entries with 1-2 dynamic pedestrians added)" based on this scheme, and tested the core performance of NaVid-SAGE and NaVILA-SAGE on this subset. The experimental results are shown in the following table:
>
> |Model|Scene Type|SR|OSR|SPL|CSR|
> |-|-|-|-|-|-|
> |NaVid-SAGE|Purely Static Scenes|0.36|0.46|0.32|0.48|
> |NaVid-SAGE|Dynamic Scenes|0.28|0.35|0.25|0.38|
> |NaVILA-SAGE|Purely Static Scenes|0.46|0.55|0.48|0.57|
> |NaVILA-SAGE|Dynamic Scenes|0.40|0.49|0.42|0.51|
>
> The experimental results clearly show that the introduction of dynamic scenes leads to a significant decline in the core metrics of both models. This fully **indicates that the dynamic scenes we constructed are effective and highly challenging**.
>
> In summary, we highly recognize your suggestion. The current static dataset serves as the foundation for constructing high-quality dynamic scenes, and the expansion of dynamic scenes is our clear follow-up direction.

---

### Author Response · Authors · 2025-11-26
**General Response**

We sincerely thank all reviewers for their detailed and constructive feedback, which has significantly improved the quality and rigor of our manuscript. **We are encouraged that the Reviewers recognized the following key merits**:

- The novelty and value of the SAGE-3D paradigm, which upgrades 3D Gaussian Splatting (3DGS) from a purely perceptual representation to an executable, semantically and physically aligned environment for embodied navigation (Reviewer **hytT**, **Wreg**, **zosw**).
- The significance of the constructed resources: InteriorGS (1K annotated 3DGS indoor scenes) and SAGE-Bench (the first 3DGS-based VLN benchmark with 2M instruction-trajectory pairs and 554k collision bodies), which fill gaps in the field (Reviewer **hytT**, **mZXf**, **zosw**).
- The paper's clear organization, accessible presentation, and structured exposition of complex concepts, making the core contributions easy to follow (Reviewer **hytT**, **Wreg**).
- The meaningful experimental design, which demonstrates the framework's advantages in rendering efficiency, model generalization, and supporting continuous navigation evaluation (Reviewer **hytT**, **Wreg**, **zosw**).

To address the concerns raised by the reviewers, we have conducted several additional experiments:

- **Explored a scalable dataset construction pipeline** that combines SceneFactor for 3D scene generation, SAI3D for instance segmentation and GPT-5 for semantic annotation. We designed four training configurations to validate the irreplaceability of manual data and the diversity gain of the automated pipeline.
- **Built a dynamic scene test subset** based on IsaacSim and AdaVLN, adding 1-2 dynamic pedestrians to test entries. We tested NaVid-SAGE and NaVILA-SAGE models, and the observed metric declines confirm the effectiveness and challenge of dynamic scenes.
- **Conducted instruction quality evaluation** including a user study with 10 participants assessing 1000 instructions across accuracy, relevance and executability dimensions, paired with n-gram diversity and category balance metrics, plus experiments on the impact of low-quality instructions.
- **Supplemented CSR metric validation** by analyzing the optimal path distribution of 100k trajectories, finding that the overlap rate between alternative paths and the A* path reaches 91.7%. We also performed 50 dual-path tests, which show CSR’s low sensitivity to alternative optimal paths.
- **Verified training stability & label quality** through five repeated trainings (convergence epochs 155-167, SR standard deviation 1.8%) and a 10k-sample blind evaluation (label error rate 0.76%), ruling out issues of training instability and label noise.
- **Performed semantics/physics ablation studies** where removing 50% of semantic annotations dropped SR to 0.32, and disabling physical collision constraints reduced SR to 0.29. These results confirm both semantic and physical information are indispensable for navigation performance.

We have also clarified the following key points:

- Explained why our model’s CR/ICP/PS metrics are lower than baselines: baselines’ poor navigation (random/single actions) leads to distorted metric values, supported by case studies.
- Clarified SAGE-Bench’s core contributions: innovations in 3DGS-Mesh Hybrid Representation, hierarchical causal instructions, and continuity evaluation metrics.
- Addressed minor issues: corrected Introduction typo, renamed "Nogoal-Nav" to "visual exploration task", and optimized table readability by bolding top results.

We greatly appreciate the reviewers’ recognition of both the originality and technical soundness of our work, and especially their engagement, which enabled us to address targeted concerns and present a more compelling and complete manuscript.

Best regards, Authors

---

### Meta-Review · Area_Chair_FZdo · 2025-12-23

**Summary:**

The paper proposes SAGE-Bench, a benchmark for embodied navigation in a continuous setting where 3D gaussian splats (3DGS) are used as the 3D representation. As the traditional 3DGS mainly captures visual appearance, it is augmented with semantics and phyical information (e.g. collision meshes) to form the SAGE-3D representation.  The paper also introduces Interior3D, a dataset of 1K indoor 3DGS scenes with over 554K object instances over 755 categories.   SAGE-Bench is built on top of Interior3D and targets vision-language navigation (VLN) and visual exploration tasks, with VLN instructions of varying complexity (from low-level to high level).   Experiments compared different models on the proposed SAGE-Bench, and demonstrately usefulness of models trained on SAGE-Bench on alternative VLN datasets (e.g. R2R for VLN-CE).

Overall, reviewers were all positive on this work, and recommend acceptance.  The AC note that while prior work has used 3DGS for navigation tasks, this is the first work to provide a dataset and benchmark so that others can also easily use 3DGS for embodied navigation.

The authors provided a response on Nov 26, and there was no followup responses from the reviewers.  The AC believes most of the reviewer concerns were addressed with thorough experiments.

The manuscript did not clearly indicate updates, but did look to address some of the reviewer comments. The AC could not find the additional experiments provided for the author response in the revised manuscript.  The AC recommend the authors adds the additional experiments and information in the appendix.

Overall, given the initial scores from reviewers, the AC believe reviewers would be positive on this work and recommends acceptances.

**Reviewer Concerns:**

Reviewers noted the following weaknesses
1. Limitation of dataset
   a. Dataset construction required extensive manual annotation which may not be easily replicated / scaled [hytT, Wreg]
   b. Dataset is only for static meshes and lacks dynamic objects [hytT]
2. Question about quality of MLLM generated instructions [Wreg]
3. Ablation to show the importance of added semantics and physical information [mZXf]
4. There are some anomalies with the experiments that were not sufficient discussed [Wreg] including:
   a. questions about why some metrics are lower for SAGE-based models than baselines [Wreg, mZXf]
   b. questions about why models trained on SAGE-Bench are "harder to converge" [Wreg]
5. Question about design choices for top-down map generation, metric definition [Wreg]
6. Question about memory and computational cost of added parameters and collision bodies [zosw]
7. Small presentation issues [Wreg, zosw]

The AC note that the authors provided a very thorough response to the reviewer comments, including additional experiments investigating automated dataset construction (1a), adding pedestrians for dynamic scenes (1b), user study to assess quality of generated instruction (2), ablations to show important of added semantics and physical information (3), as well as information / discussion / experiments to address 4-6.  The paper was also revised to fix the small presentation issues identified by the reviewer.

**Reviewer Scores:**

Reviewers are overall positive on this work with two marginal accepts (hytT,mZXf) and two accepts (Wreg,zosw).  As the author response was thorough, reviewers will likely have increased their scores (perhaps after the authors actually revised the manuscript to include the additional experiments).

---

### Decision · Program_Chairs · 2026-01-26

Accept (Poster)